# Used economy market insight: Sailboat industry pricing mechanism and regional effects

Zhanni Huang[1,2], Hansheng Hu[3], Di Wu[2,4]*

1 School of Economics, The University of Edinburgh, Edinburgh, Scotland, 2 School of Statistics and Mathematics, Yunnan University of Finance and Economics, Kunming, Yunnan, China, 3 International Business School, Yunnan University of Finance and Economics, Kunming, Yunnan, China, 4 National Institute of Development Administration, International College, Bangkok, Thailand

* di.w@stu.nida.ac.th

**Data Availability Statement:** All relevant data are within the manuscript and its SupportingInformation files. The data can be accessed and downloaded from the following link: https://doi.org/10.7910/DVN/FGHFMM.

## Abstract

With the popularity of circular economy around the world, transactions in the second-hand sailboat market are extremely active. Determining pricing strategies and exploring their regional effects is a blank area of existing research and has important practical and statistical significance. Therefore, this article uses the random forest model and XGBoost algorithm to identify core price indicators, and uses an innovative rolling NAR dynamic neural network model to simulate and predict second-hand sailboat price data. On this basis, we also constructed a regional effect multi-level model (RE-MLM) from three levels: geography, economy and country to clarify the impact of geographical areas on sailboat prices. The research results show that, first of all, the price of second-hand sailboats fluctuates greatly, and the predicted value better reflects the overall average price level. Secondly, there are significant regional differences in price levels across regions, economies and ethnic groups. Therefore, the price of second-hand sailboats is affected by many factors and has obvious regional effects. In addition, the model evaluation results show that the model constructed in this study has good accuracy, validity, portability and versatility, and can be extended to price simulation and regional analysis of different markets in different regions.

## Introduction

With the popularity of circular economy around the world, the principles of "reduction, resource utilization, and reuse" in the circular economy have regained widespread attention [1]. Especially as the concept of green and low-carbon consumption gradually becomes more and more popular, buying second-hand goods has gradually become the first choice of consumers [2]. As an important part of the circular economy, second-hand items are a key driving force for the realization of a circular economy, and their development direction has also attracted much attention [3,4]. The 2023 Government Work Report clearly states: We must develop a circular economy and promote the conservation and intensive use of resources [5]. It can be seen that the development of the second-hand commodity trading market has a

**Funding:** The author(s) received no specific funding for this work.

**Competing interests:** The authors have declared that no competing interests exist.

significant economic role in promoting green lifestyles, rationalization of consumer spending, and sustainable economic development. However, the current trading behavior of second-hand goods still has problems such as imperfect regulations and insufficient popularity [6]. This is inseparable from the "arbitrary" pricing mechanism of second-hand goods, especially the unreasonable pricing mechanism of second-hand luxury goods. Therefore, realizing scientific pricing of second-hand goods is an important prerequisite for promoting the development of circular economy and achieving sustainable development.

As an ancient water vehicle, sailing boats originated in the Netherlands in the 16th and 17th centuries and have a history of more than 5,000 years [7]. However, in the public eye, sailing has always been considered a luxury product because of its high price, mooring fees and maintenance costs [8]. However, with the rapid development of emerging industries such as sailing events, sailing culture, and sailing tourism, the global interest and demand for sailing sports continues to increase [7], which makes the second-hand sailboat market extremely active. According to relevant data, since 2018, the size of the global second-hand ship trading market has increased year by year, and the transaction volume is shown in Fig 1. It can be seen that the sailboat industry is becoming increasingly mature and developing extremely rapidly in the global second-hand market [9]. Therefore, potential market dividends are attracting more brokers to dig deeper into the second-hand sailboat market. So, how to scientifically formulate a second-hand sailboat pricing mechanism, or clarify the influencing factors of second-hand sailboat pricing, is the key to promoting the healthy development of the second-hand sailboat industry.

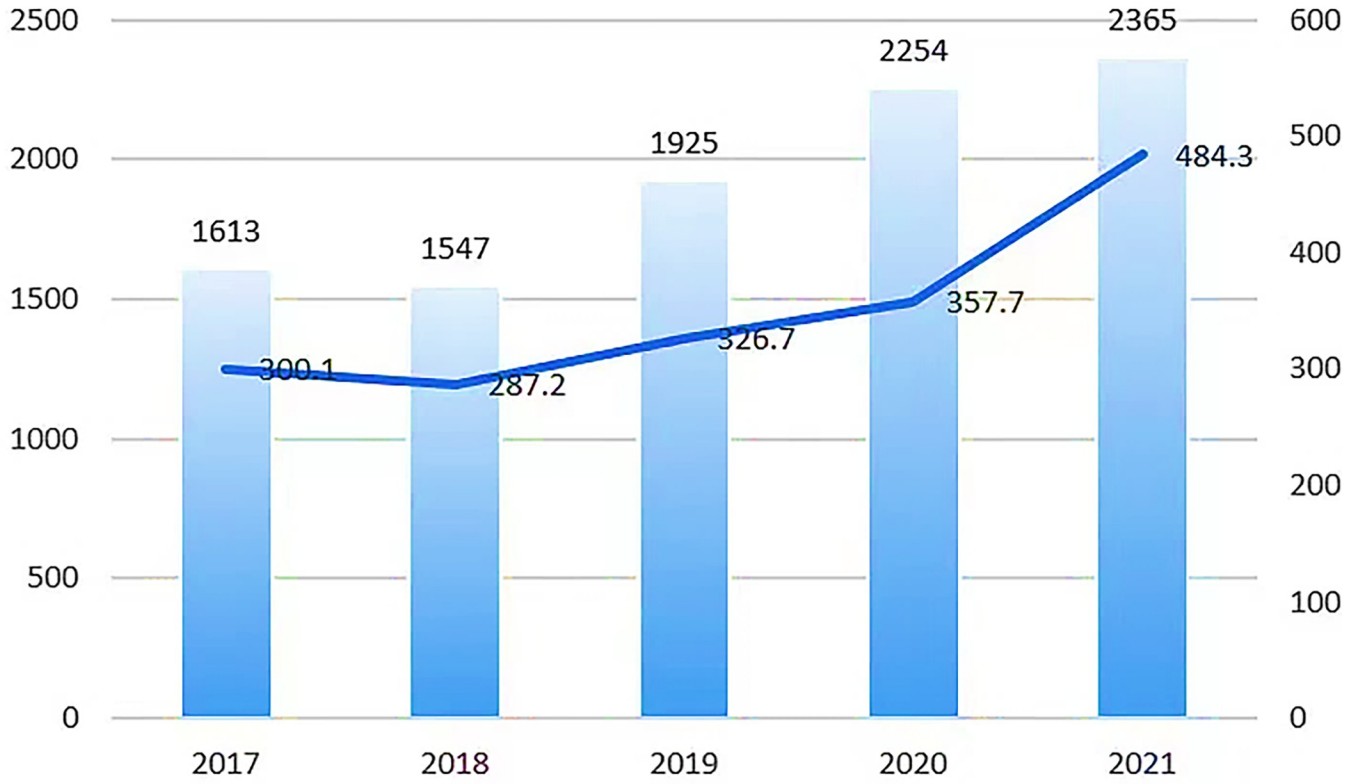

**Fig 1. Global used ship transaction size.**

Through in-depth exploration of relevant literature and the second-hand sailboat market, this study found that second-hand sailboat pricing has clear influencing factors and regional effects. However, clarifying pricing strategies and exploring their regional effects are still gaps in the current research field, which has very important theoretical and practical significance. Therefore, in order to help brokers discover the pricing mechanism rules and regional effects of the second-hand sailboat market, and apply them to practice. This article aims to address the following questions by analyzing data from approximately 3,500 sailboat advertisements sold in Europe, the Caribbean and the United States in December 2020.

It is well known that the price of used sailboats is one of the focal issues of public concern, and D'Adamo, Lupi [9] mentioned that the price of sailboats can be affected by a combination of factors such as material, value and market etc. Therefore, we analyze the relationship between price and many factors based on the characteristics of sailboats, and aims to study the laws of potential pricing mechanisms in the used sailboat market.

Sailboats are particularly flourishing in near-sea countries or regions such as Europe and the Americas, but regions have large disparities in sailboat manufacturing base, production, and sales [10]; therefore, this paper hypothesizes that geographic regions have an effect on used sailboat prices and explores whether regional effects are consistent across models.

The validity of theoretical studies often needs to be validated by cases. The article applies the above model to the Hong Kong (SAR) market and simulates the regional effects of Hong Kong (SAR) on a subset of sailboats to assess the generalizability and robustness of the model.

## Literature review

As an important part of the sharing economy, the "second-hand economy" is not only conducive to spreading the concept of green consumption, but also responds to the development trend of the global circular economy [9]. According to the data from the U.S. bank KeyBank shows that the current U.S. primary market for luxury goods is worth about $300 billion, while the secondary resale market for luxury goods can be worth $600 billion. However, while the second-hand economy is booming there are still hidden dangers [11]. The whole industry lacks clear business models and industry rules, especially the pricing mechanism is vague [12]. Thus, this paper is of great relevance to the study of pricing in the used sailboat market.

Currently, there is a large amount research on the secondary market, covering topics such as trading platforms and new energy [13,14]. In terms of research methods, most of them are analyzed through data from physical secondary market surveys, while for the study of pricing decisions, different scholars use models with their own characteristics. Yeh and Lien [15] uses a hybrid approach of binomial option pricing model and Monte Carlo simulation to solve real estate development project pricing problems, while Gu, Luo [16] use a two-period dynamic pricing decision model to study a secondary trading platform, providing ideas for this paper to use an innovative rolling neural network pricing model. Based on the above analysis, this article puts forward the following research hypotheses and explanations.

**Assumption 1:** The depreciated value of a used sailboat is only affected by the year factor.

**Explanation 1:** Depreciation rate is an important indicator of the value of used goods and is influenced by multiple factors such as frequency of use, product quality etc [17]. Considering the complexity of the calculation, we uniformly use the "ten-year depreciation method" for the calculation.

**Assumption 2:** Pricing mechanisms are mainly influenced by objective factors.

**Explanation 2:** Due to the uncontrollable nature of subjective factors and considering the availability of data, we mainly base our analysis on macro factors, sailing parameters and other data.

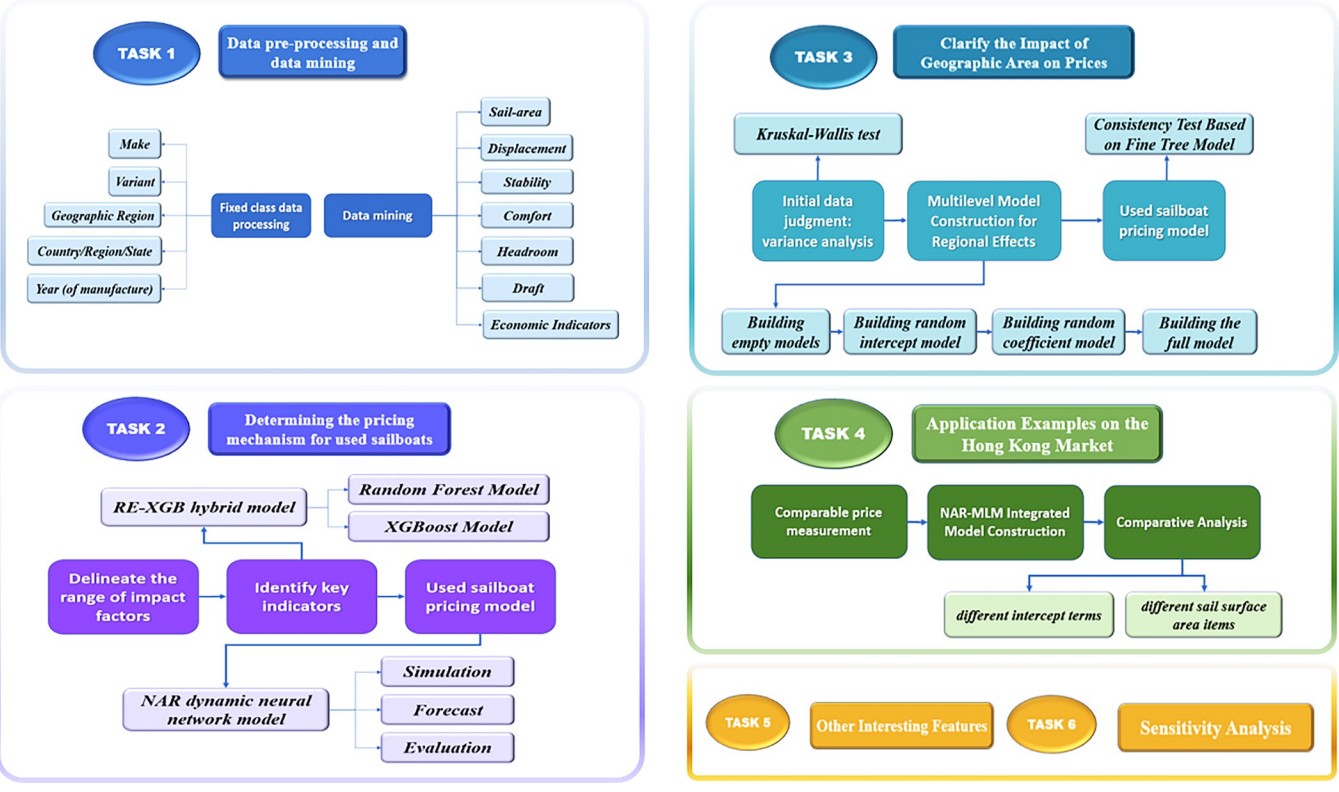

**Fig 2. Mind map of our work.**

**Assumption 3:** The full sample is a random sample and the sample size of each region is positively correlated with local market activity.

**Explanation 3:** Due to the large variation in the data sample size collected in different regions, simplifications were made based on probabilistic theory.

In addition, the modeling process of this article is shown in Fig 2.

## Data mining and preprocessing

### Web data crawling

In order to measure the factors influencing the pricing of sailboats, we added detailed descriptions of specific sailboat characteristics, as well as economic data by region, in addition to analyses from the perspective of *Make, Variant, Length (in feet), Geographic Region, Country/Region/State, Listing Price (in US dollars)*, and *Year (of manufacture)*.

Since there is less data related to sailboats, we take full advantage of big data and use web data crawling to crawl the sailboat specification parameters through the SciPy library in Python, the specific operation process is shown in Fig 3.

Based on the data available in the web and with reference to existing studies, Drobetz, Ehlert [10] proposes that equipment and materials of sailing boats cause cost differences, including sail surface, mast, engine, displacement, locks, etc. An, Yu [7] states that the overall performance of sailing boats also causes differences in the selling price of sailing boats, with performance including stability, comfort, headroom, electronics, etc. Therefore, our additional data indicators include *Sail-area, Displacement, Stability, Comfort, Headroom, Draft* and *Economic Indicators*.

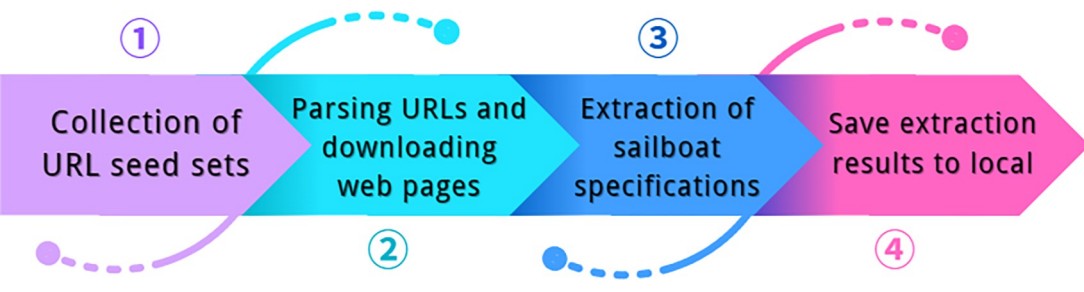

**Fig 3. Web crawler flow chart.**

At the same time, we also crawled relevant market data in Hong Kong (SAR) to analyze the largest trading platform for yachts and boats in Hong Kong and other data as an important basis for the analysis. The main websites involved in the data mining web crawl are listed in Table 1.

## Data preprocessing

In order to avoid errors due to the endogeneity problem of the data, this paper adopts a positioning model and preprocesses the missing values such as interpolation supplementation and type name correction. Since the data are nonparametric, i.e., they include both quantitative and definite type data, we first quantified the definite type data to prepare for the subsequent research analysis.

**Fixed class based data processing.** Based on the above-mentioned study by Gu, Luo [16] we found that the definite class indicators such as *Make*, *Variant*, *Geographic Region*, *Country/ Region/State*, and *Year* (of manufacture) all have a significant impact on the price of used products [18], therefore, we quantified the assignment by taking a single sailboat as an example, with a full sample of 2346, and considered the sample as a random sample.

**Make** →Assign values according to the manufacturer's global brand awareness.

There were 61 single-unit sailboat manufacturers in the full sample, and we assigned quantitative values based on the percentage of individual frequencies, which are shown in Table 2.

According to the survey, the French Beneteau Group is known as "the world's largest sailboat manufacturer", and the famous French boat brand Jeanneau and German Bavaria are also known worldwide.

**Variant** →Assigning values according to network hotness.

By crawling the website for sailboat models, the model hotness was used to reflect the level of consumer preference. Among the global sailboat models, Oceanis, Impression, Cruiser, Sun

**Table 1. Data and database websites.**

| Database Name | Database Websites |
|---|---|
| Sailboat Parameters Indicators | http://www.eworldship.com/ |
| | https://www.cnss.com.cn/ |
| | http://www.ishipoffshore.com/ |
| | https://iacs.org.uk/ship-company-data/ |
| Hong Kong (SAR) Sailboat Data | http://www.boatuc.com.hk/en/index.asp |
| | https://hkmb.hktdc.com/ |
| | http://www.luxboating.com |
| Hong Kong (SAR) Industry Data | http://www.censtatd.gov.hk/ |
| | http://www.hkma.gov.hk/eng/index.shtml |

**Table 2. Quantitative assignment of manufacturer data.**

| Make | Jeanneau | Beneteau | Bavaria | Hanse | Dufour |
|---|---|---|---|---|---|
| Frequency / Percentage | 503/ 21.441% | 497/ 21.185% | 328/ 13.981% | 177/ 7.545% | 161/ 6.863% |

Odyssey, and First series appear more frequently on the web and receive most consumers' attention and preference.

**Geographic Region** →Assign values according to the market share of sailboat manufacturing.

By reviewing the information, the French Beneteau Group has more than one-third of the global market share, and there are other well-known brands in Europe such as Jeanneau in France and Bavaria in Germany [10]. Since the overall market share is higher in Europe, followed by the United States and Caribbean, we assign 10 to Europe, 8 to the United States, and 6 to Caribbean.

**Country/Region/State** →Assigning values according to the economic level of each region.

The 72 countries or regions involved in the overall sample are ranked and assigned according to the economic level (GDP index) of each country.

**Year (of manufacture)** →Adjust the year data for the time of manufacture to the depreciation rate data for used sailboats.

We extend the depreciation rate of industrial used cars to the calculation of the depreciation rate of used sailboats by using the "ten-year depreciation method" [17]. Using 2019 as the base year (with a depreciation rate of 1), the depreciated values for each year are shown in Table 3.

## Parameter symbols

The model parameters involved in this paper are shown in Table 4.

## Determining the pricing mechanism for used sailboats

## Method description

This paper addresses the problem of used sailboat pricing mechanism through the following three steps: firstly, we delineate the range of price influencing factors based on literature and correlation analysis, then we use the RE-XGB hybrid model to determine the main indicators, and finally, we innovatively construct an improved NAR dynamic neural network model to simulate and predict used sailboat prices.

**Correlation analysis of influencing factors.** The pricing mechanism of used sailboats can be influenced by many factors. Based on the aforementioned literature, we summarized 11 indicators (*MFR, VAR, LEN, RGN, SSA, STB, DPM, CNTRY, CMFT, HDR, DRAFT*). In order

**Table 3. Depreciated value of used sailboats by year.**

| YEAR | Depreciated value | YEAR | Depreciated value |
|---|---|---|---|
| 2019 | 1 | 2011 | 0.404 |
| 2018 | 0.85 | 2010 | 0.3838 |
| 2017 | 0.7225 | 2009 | 0.3646 |
| 2016 | 0.6141 | 2008 | 0.3463 |
| 2015 | 0.5527 | 2007 | 0.3289 |
| 2014 | 0.4974 | 2006 | 0.3125 |
| 2013 | 0.4477 | 2005 | 0.2969 |
| 2012 | 0.4253 | . . . | . . . |

**Table 4. Model parameters.**

| Symbols | Definition | Symbols | Definition |
|---------|-----------|---------|-----------|
| PRICE | The advertised price of the boat. | MFR | The manufacturer of the boat. |
| VAR | The name of model. | LEN | The length of the boat in feet. |
| RGN | The geographic region. | CNTRY | The specific country. |
| SSA | The total surface area of the sails of a boat when fully raised. | CMFT | The comfort of the boat. |
| STB | The stability of the boat. | HDR | The height available to stand up in the cabin. |
| DPM | The weight of the volume of water displaced by a boat. | DRAFT | The minimum depth of water required to float a boat without touching the bottom. |

to get a more intuitive preliminary understanding of the factors influencing prices, we first analyze the correlation between each indicator and prices.

As seen in Fig 4, the price of monohull sailboats is better correlated with *DRAFT*, *MFR*, *LEN*, *SSA*, *DPM*, *STB*, *CMFT*, *HDR*; while in catamaran sailboats, the price is equally correlated with *LEN*, *SSA*, *DPM*, *STB*, *CMFT*, *HDR*, *DRAFT*. Thus, we initially selected the above eight indicators for the subsequent analysis.

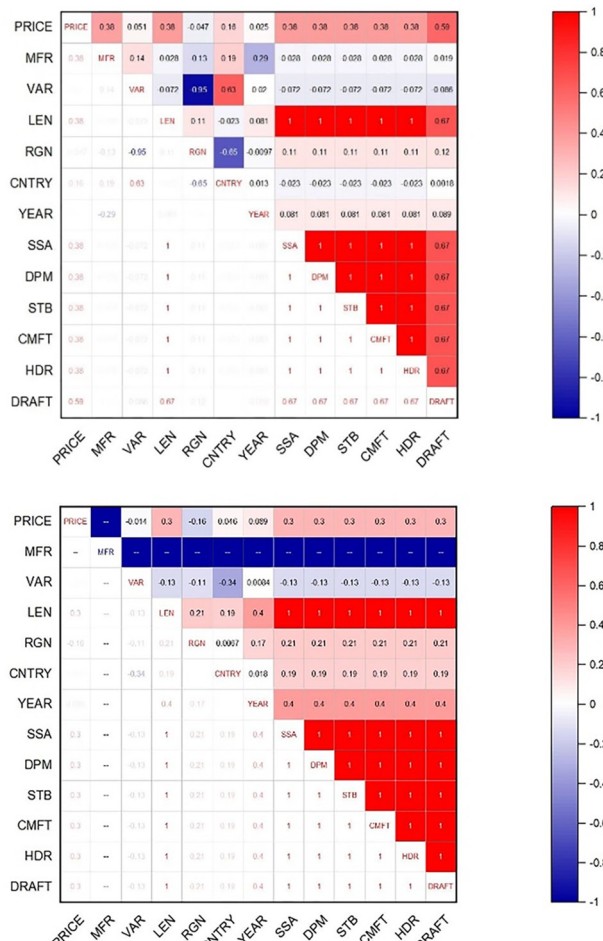

**Fig 4.** (a). Correlation analysis of influencing factors. (b). Correlation analysis of influencing factors.

## Selection of important indicators based on RF-XGB hybrid model

In order to analyze the degree of influence of each factor more precisely, we further used the random forest model and XGBoost model to screen the main indicators, and the training set data to build a regression model and calculate the feature importance.

**Model metrics measurements.** The RF-XGB hybrid model refers to an integrated measurement model consisting of a random forest model and an XGBoost model. As a result, we take the example of a single sailboat, and the following Fig 5 shows the importance of the indicator characteristics calculated by the model.

From the proportion of feature importance in the figure, we can see that four indicators, *DRAFT*, *MFR*, *VAR* and *YEAR* can explain nearly 90.2% of the information in monohull sailing, and similarly, four indicators, *VAR*, *YEAR*, *CNTRY* and *RGN* can explain nearly 90% of the information in catamaran sailing. Therefore, the subsequent simulations and predictions using these four indicators can effectively reduce the number of indicators and have high validity.

**Model evaluation.** Based on the construction of the RF-XGB hybrid model described above, we evaluate the results of the training set, as shown in Table 5.

The table shows the evaluated values of mean square error, root mean square error, mean absolute error, mean absolute percentage error and goodness of fit, which shows the higher accuracy of the model.

## Used sailboat pricing NAR dynamic model

**NAR dynamic neural network model construction.** NAR (Nonlinear Auto-Regressive) neural network is called nonlinear autoregressive model, which is a dynamic neural network. The output of the model at each moment is based on the synthesis of the dynamic results of the previous system at the current moment, i.e., it has the function of feedback and memory [18].

Compared with BP neural network, NAR neural network has the characteristics of both dynamic and complete system information, which not only inherits the advantages of traditional time series model, but also has better adaptability and prediction effect for nonlinear data [19]. In this paper, the NAR dynamic neural network model will be applied to simulate and predict the price of used sailboats.

The model of NAR neural network can be described as:

$$Y(t) = f(y(t-1), y(t-2), y(t-3), \cdots, y(t-d)) \tag{1}$$

where $Y(t)$ is the value of the variable at the current moment; $y(t-1), y(t-2), y(t-3), \cdots, y(t-d)$ is the value of the variable at the historical moment; $d$ is the delay order.

In the Fig 6, *Input* denotes the input of the neural network; $1:d$ in the hidden layer is the delay order, which indicates the use of $d$ points before a point in the time series to predict the value of that point; $w$ is the connection power and $b$ is the threshold.

The individual neuron outputs can be expressed as:

$$H_i = f(\sum_{i=1}^{n} w_{ij} y_i + b_j) \tag{2}$$

where $f$ is the activation function; $w_{ij}$ is the connection weight between the $i$ output delay signal and the $j$ neuron in the hidden layer.

**Used sailboat NAR dynamic pricing analysis.** After reviewing a large amount of literature and inspired by scholars such as Gu, Luo [16], Kumar and Kumar [18], this paper

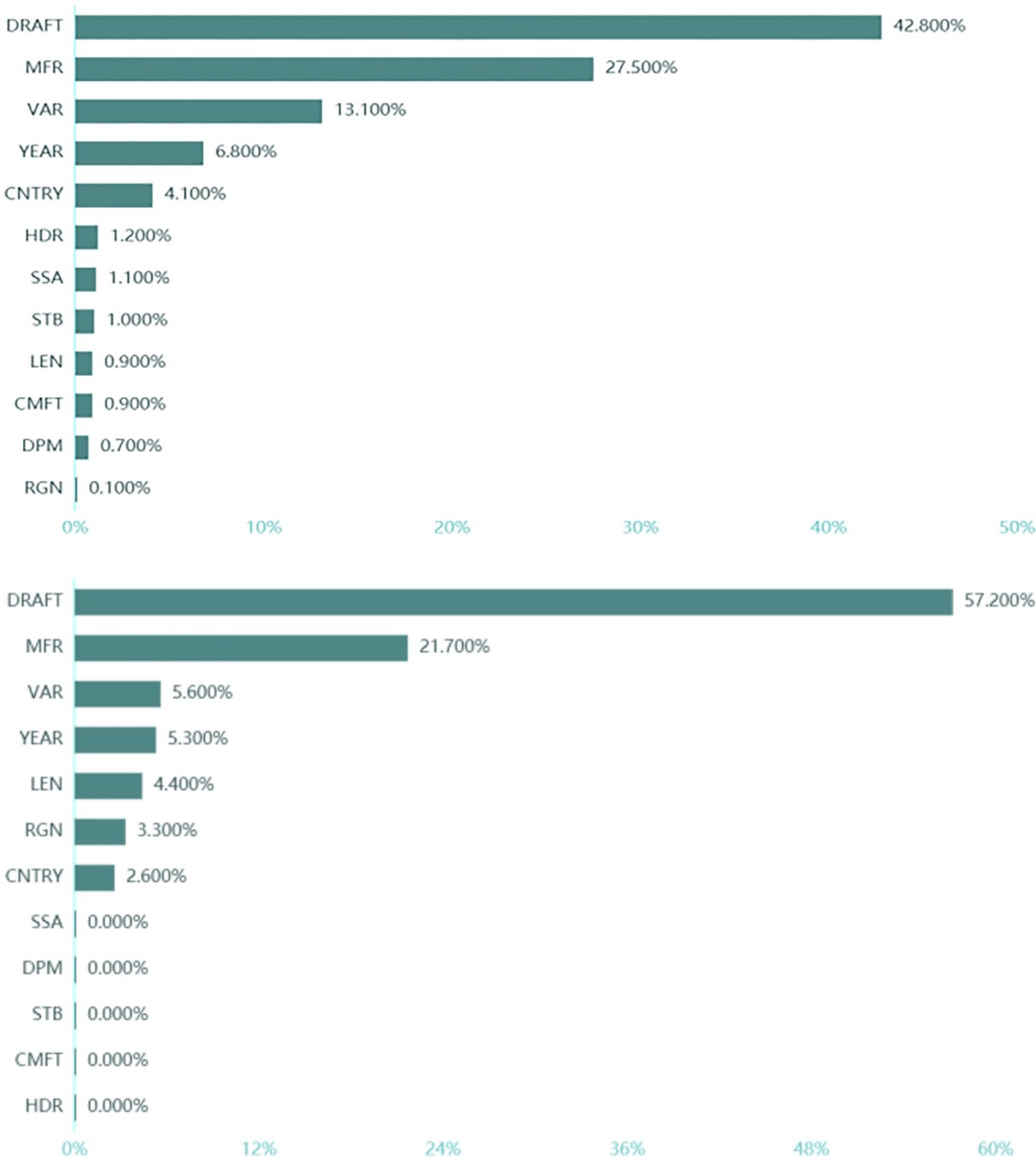

**Fig 5.** (a). Importance of single sailboat characteristics for RF-XGB hybrid model. (b). Importance of single sailboat characteristics for RF-XGB hybrid model.

**Table 5. Important indicators selection model evaluation.**

| Algorithm | Dataset | MSE | RMSE | MAE | MAPE | $R^2$ |
|---|---|---|---|---|---|---|
| Random forest regression | Training set | 2786082583.181 | 52783.355 | 38320.137 | 19.756 | 0.802 |
| XGBoost returns | Training set | 275703739.120 | 16604.329 | 9879.743 | 6.087 | 0.980 |

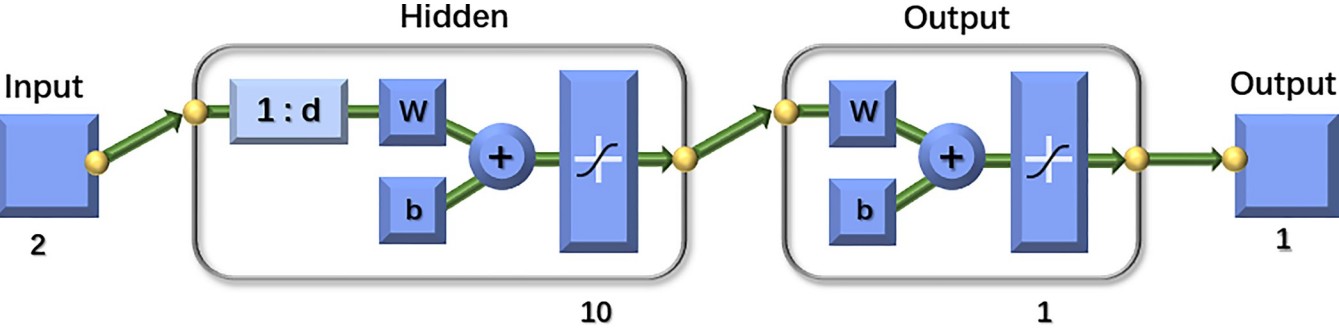

**Fig 6. NAR dynamic neural network model.**

proposes to construct the aforementioned NAR dynamic neural network model to be applied in the study of the pricing mechanism of used sailboats.

## Deep learning price simulation

Based on the above model theory, we set the autoregressive order lag to 2 and set the number of hidden layer neurons to 10 in the created network. Meanwhile, in order to avoid overfitting, we divided the proportion of training, testing and for prediction test data to 70%, 15% and 15% respectively. The results of the price simulation of specific used sailboats are shown below.

From the Fig 7, we can see that the simulation of the deep learning of the used sailboat pricing model is good, and the target value is more consistent with the output value, which is basically in line with the overall data trend, and the error is small overall, which is controlled within 5%.

## Rolling dynamic price forecasts

Considering that prices in the secondary market are prone to fluctuations due to market supply and demand [11], we innovatively improve the delay order to a rolling dynamic forecasting

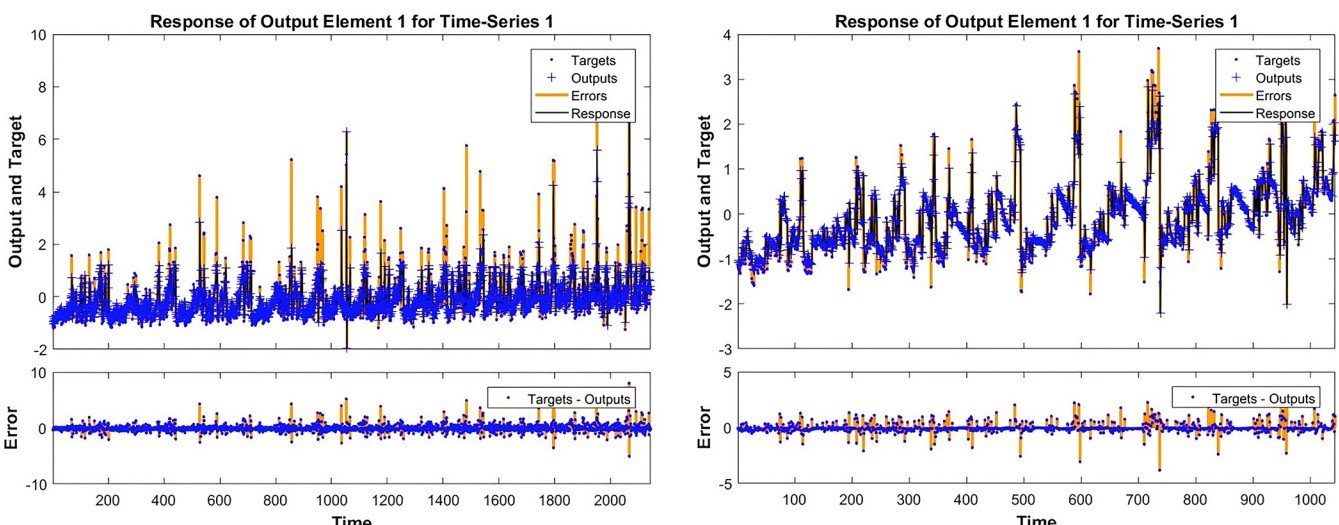

**Fig 7. Catamaran and monohull sailboat price simulation effect.**

model that predicts prices in the subsequent period by rolling the price levels of the previous periods.

In the double-hull sailboat NAR dynamic forecasting model, rolling forecasts are made starting from the 101st period, i.e., the data from the 1st to the 100th period are used to predict the pricing level of the 101st period, the data from the 2nd to the 101st period predict the pricing level of the 102nd period and so on. Similarly, in the single-unit sailboat NAR dynamic forecasting model, due to the large sample size of the single-unit, we set the forecasts to start rolling every 200 issues, and the results of each forecast will have an impact on the later results on a rolling basis, which is an innovative aspect of the model.

The specific prediction results are shown in the Fig 8 below.

Fig 8. Comparison of true and predicted values for double hull and monohull sailboats. From Fig 8, we find that the real price of used sailboats is characterized by strong up and down fluctuations, while the forecast value can better reflect the overall price trend of used sailboat prices, making the real price basically fluctuate up and down around the forecast value, which is a good reference value for predicting the average level of future prices.

**Model accuracy assessment.**   In assessing the accuracy of the model, autocorrelation test, partial autocorrelation test, and error analysis are performed in this paper and presented in the following form through visualization. The specific results are shown in the Figs 9–11 below.

It is obvious from the Figs 9 and 10 that most of the samples have no autocorrelation and only a few samples have a smaller degree of autocorrelation and are within the confidence interval; therefore, we consider the overall effect of the model to be good and valid.

As seen from the Fig 11, most of the sample errors are concentrated near 0. Only a very small number of values have error levels with absolute values greater than 1, indicating that the overall error of the model is small and manageable.

## Clarify the impact of geographic area on prices

**Kruskal-Wallis test for multiple independent samples.**   To gain preliminary insight into the regional data, we first conducted an analysis of variance, using a multiple independent samples Kruskal-Wallis test, and prepared for subsequent regional effects analysis.

Since the PRICE sample N < 5000, the S-W test was used and the significance P-value was 0.000***, which presents significance at the level and rejects the original hypothesis, so the data do not satisfy the normal distribution and can be subjected to multiple independent samples Kruskal-Wallis. test results are as follows.

As can be seen from the data in the Table 6, the test result p-value is 0.000***<0.05, thus the statistical result is significant and there are significant differences in PRICE between the different geographic subdivisions, with significant price differentiation across the 3 broad geographic locations. Thus, it also provides the basis for further construction of specific regional effect models.

## Multilevel model construction for regional effects

Multilevel models are powerful statistical analysis tools for dealing with structural data and have received extensive attention from researchers worldwide in recent years [20], while their application to regional effects in secondary markets is currently a gap area. Therefore, this paper proposes to construct a multilevel model to explain the effect of regional factors on secondary market prices.

**MLM basic model theory.**   The multilevel model is a statistical analysis method based on hierarchical data [21], and the analysis of hierarchical structure and k-level units is the core of

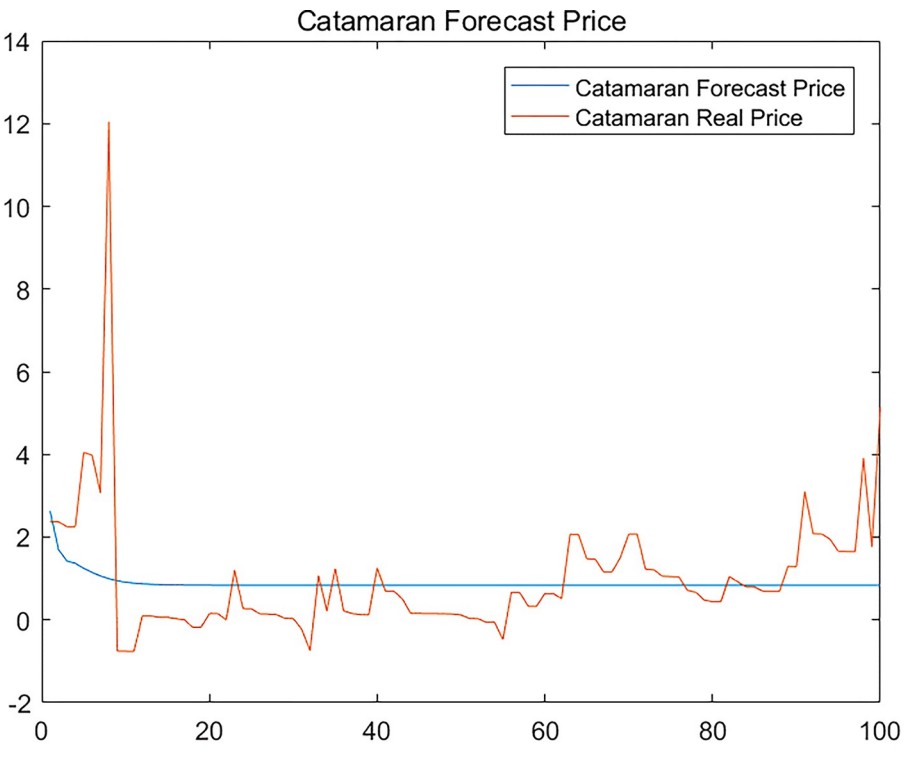

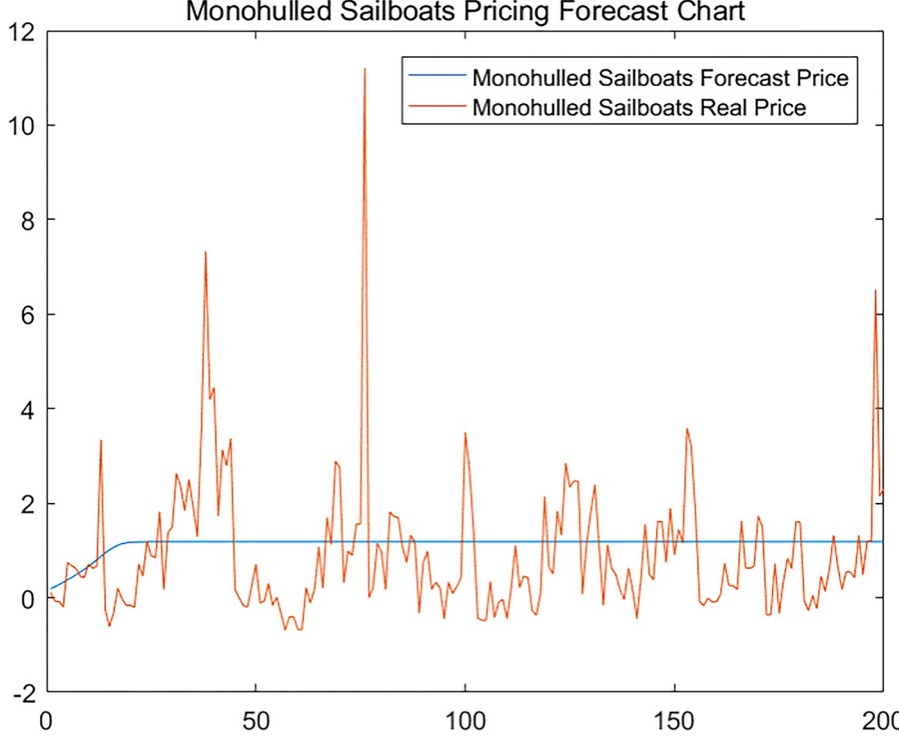

**Fig 8.** (a). Comparison of true and predicted values. (b). Comparison of true and predicted values.

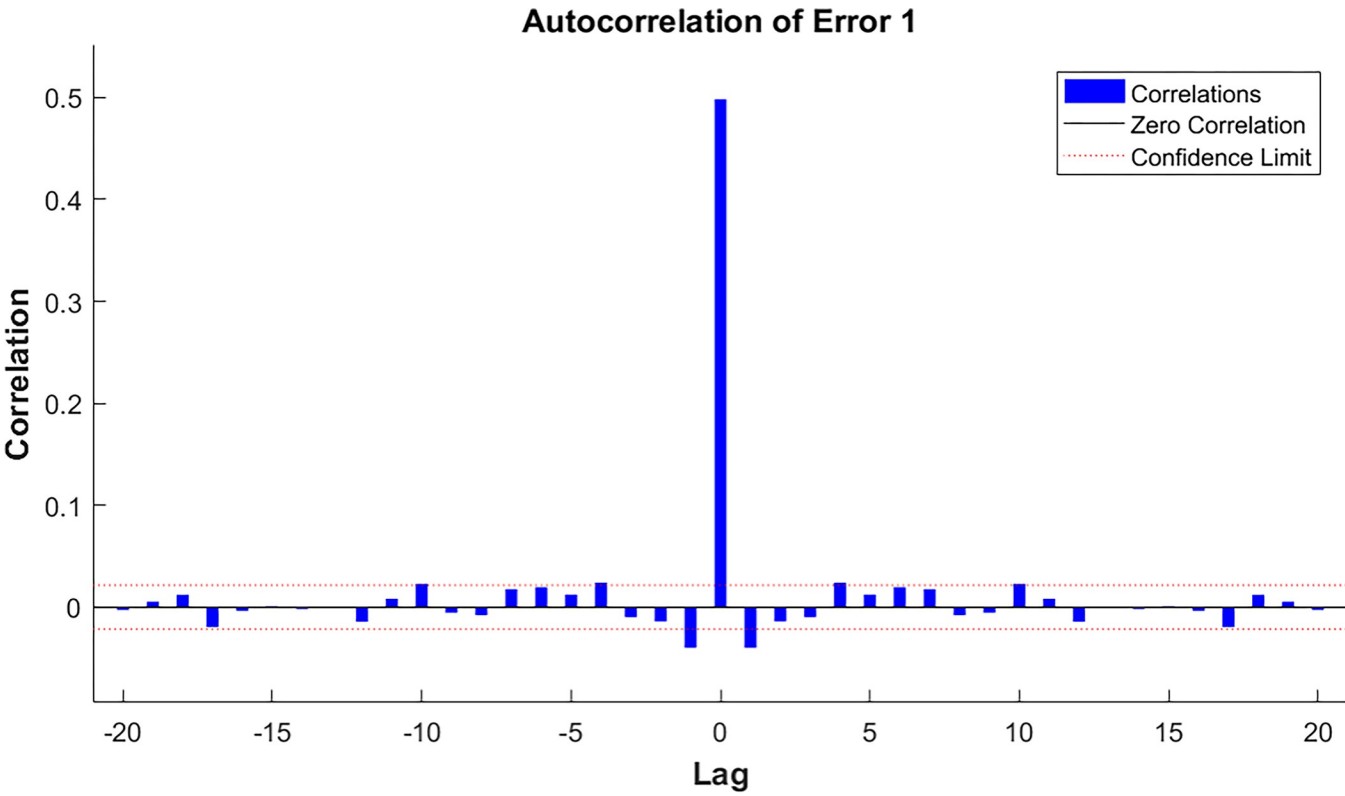

**Fig 9. Monohull sailboat autocorrelation.**

the model. The construction steps of the multi-level model are shown in Fig 12, and then we take the two-level model as an example for specific decomposition.

**Step 1: Construct the null model.** The empty model contains only random intercept terms and does not contain any independent variables.

$$y_{ij} = \alpha_0 + \mu_{0j} + \varepsilon_{ij} \tag{3}$$

$$\mu_{0j} \sim N(0, \sigma_{\mu 0}^2) \tag{4}$$

$$\varepsilon_{ij} \sim N(0, \sigma_{\varepsilon}^2) \tag{5}$$

where $i$ is the level 1 unit; $j$ is the level 2 unit to which the level 1 unit belongs; $\alpha_0$ is the total average of $y_{ij}$, $\mu_{0j}$ is the residuals at level 2, representing the between-group variation; $\varepsilon_{ij}$ is the residuals at level 1 and represents the within-group variation.

**Step 2: Construct a random intercept model.** The random intercept model is based on the above null model with a level 1 variable and the independent variable is a fixed effect.

$$y_{ij} = \alpha_0 + \alpha_1 x_{1ij} + \mu_{0j} + \varepsilon_{ij} \tag{6}$$

where $x_{1ij}$ is the level 1 independent variable added and $\alpha_1$ is a fixed effect.

**Step 3: Construct a random coefficient model.** The random coefficient model considers the random effects of the independent variables on the basis of the random intercept model.

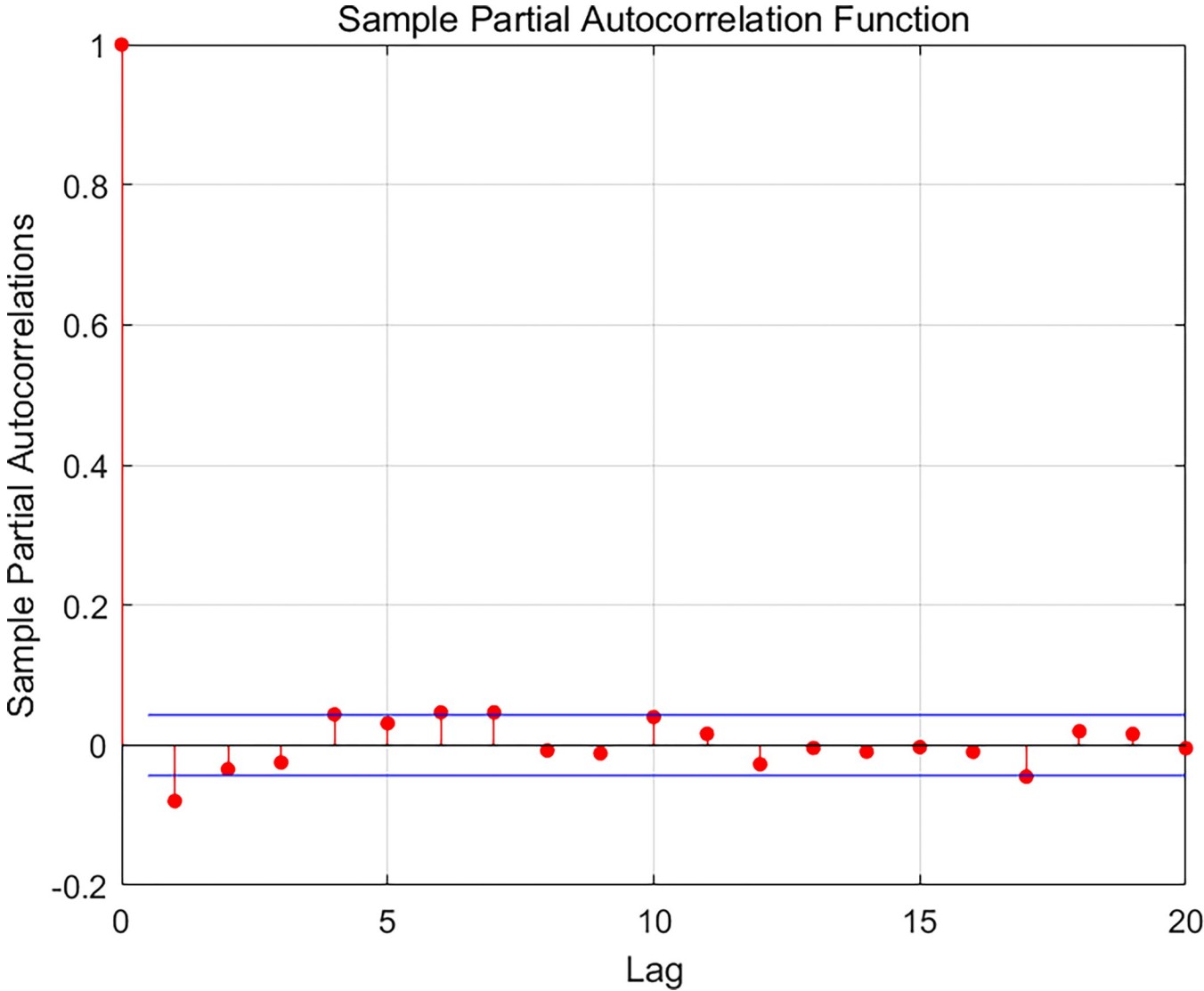

**Fig 10. Monohull sailboat bias related.**

The random coefficient model is expressed as:

$$y_{ij} = \alpha_0 + \alpha_{1j}x_{1ij} + \mu_{0j} + \varepsilon_{ij} \tag{7}$$

$$\alpha_{1j} = \alpha_1 + \mu_{1j} \tag{8}$$

$$\mathrm{var}\begin{bmatrix} \mu_{0j} \\ \mu_{1j} \end{bmatrix} = \begin{bmatrix} \sigma_{\mu0}^2 \\ \sigma_{\mu01}^2 & \sigma_{\mu1}^2 \end{bmatrix} \tag{9}$$

Where $\alpha_{1j}$ is the average effect of $x_{1ij}$ on $y_{ij}$.

**Step 4: Construct the full model.** The full model is a random coefficient model with further independent variables at level 2. Assuming that only one independent variable at level 2 is

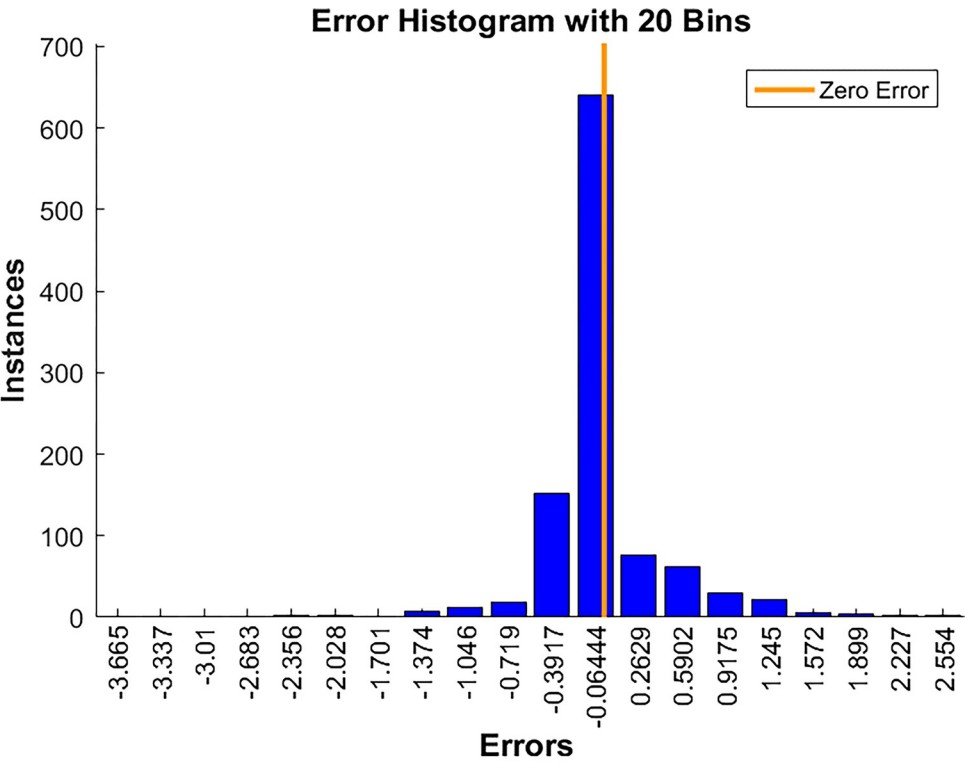

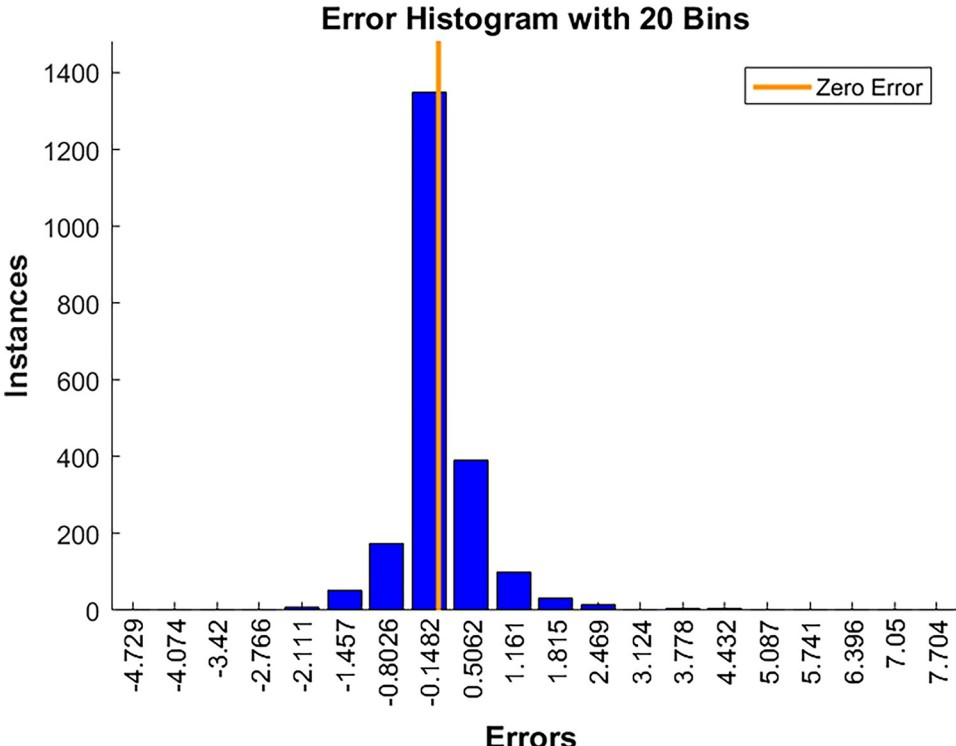

**Fig 11.** (a). Histogram of errors for double and monohull sailboats. (b). Histogram of errors for double and monohull sailboats.

**Table 6. Kruskal-Wallis test result.**

| Items | S-W test | Variables | S.D | t | P | Cohen's f-value |
|---|---|---|---|---|---|---|
| PRICE | 0.837(0.000***) | 1 | 198858 | 57.707 | 0.000*** | 0.006 |
| | | 2 | 163456.507 | | | |
| | | 3 | 215420.463 | | | |
| | | Total | 198247.117 | | | |

included, the full model is expressed as:

$$y_{ij} = \alpha_0 + \alpha_{1j}x_{1ij} + \alpha_2 w_{2j} + \mu_{0j} + \varepsilon_{ij} \tag{10}$$

where $w_2$ is the level 2 independent variable added and $\alpha_2$ is the fixed effect of $w_2$ on $y_{ij}$.

The above is the general idea of constructing a two-level model (see Figs 12 and 13), which can be extended to models at three levels and above, which happens to be applicable to the regional effects of the used sailboat market.

**RE-MLM multi-level model of regional effects.** Based on the above theory, we construct the RE-MLM model for the regional characteristics of the used sailboat market, and divide the

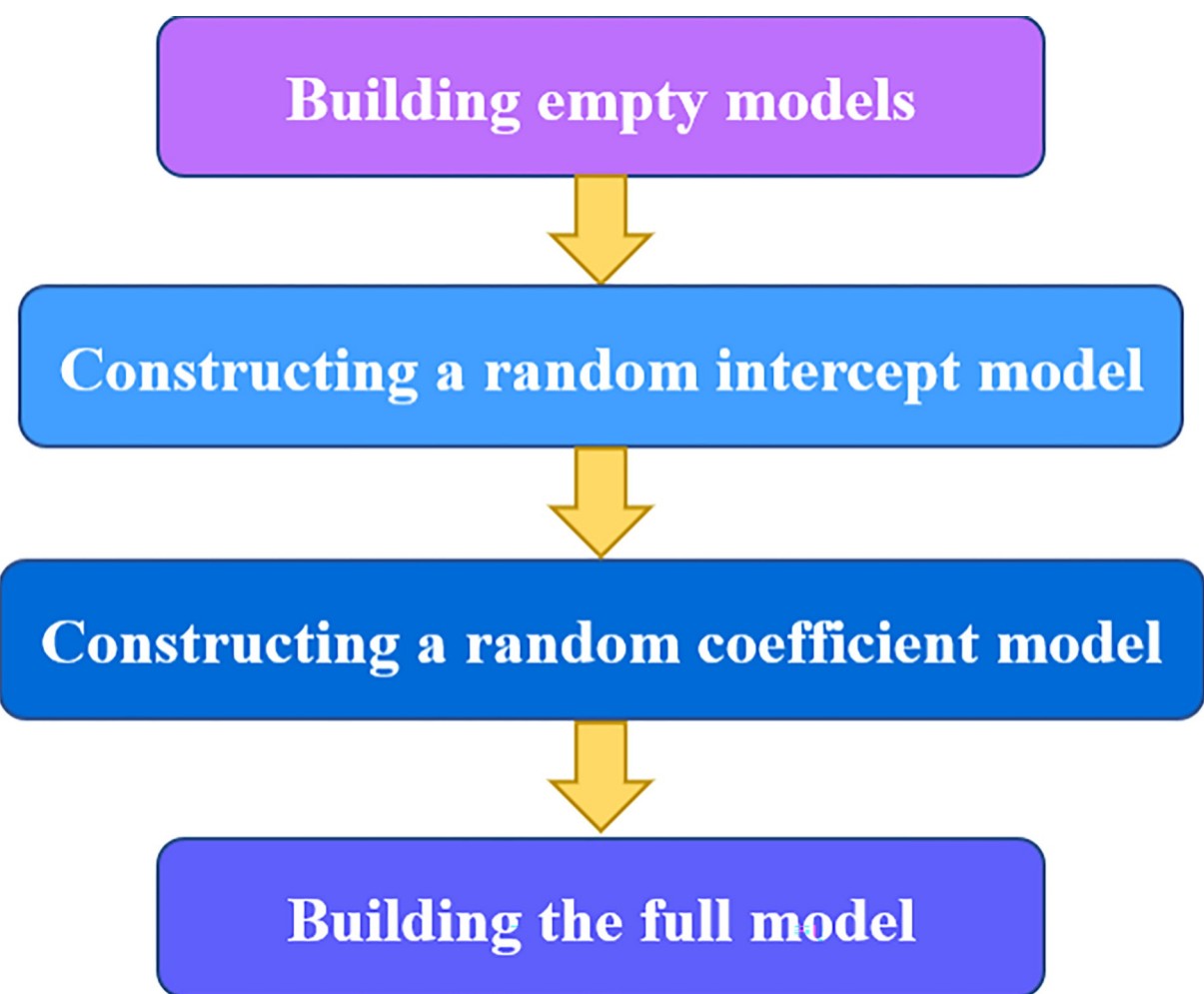

**Fig 12. Model building process.**

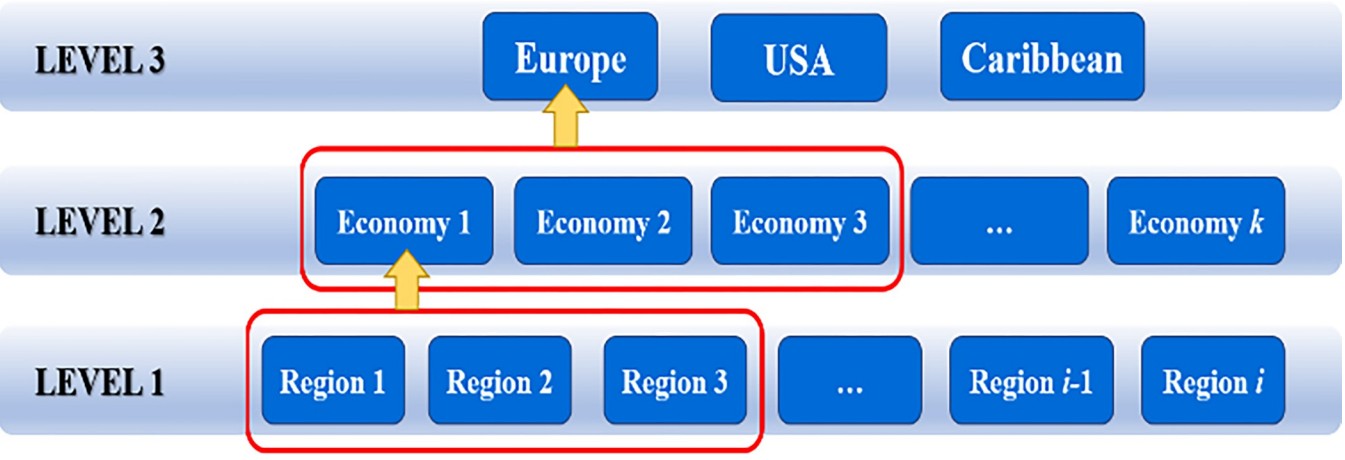

**Fig 13. Three-level model structure.**

regional structure data into three horizontal levels, and the structure schematic is shown in Fig 14.

Horizontal level division basis

**Level 3**: Europe, USA, Caribbean according to the global geographical scope of the region.

**Level 2**: According to the national economic accounting level of each country in the geographical range area, it is divided into Economy $k$.

**Level 1**: The number of advertising sample placements of used sailboats in each region within each economic level, divided into Region $i$.

Full model of the used sailboat market

The three-level model of the regional effects is constructed as follows:

$$y_{ij} = \alpha_0 + \alpha_{1j}x_{1ij} + \alpha_2 w_{2j} + \alpha_3 w_{3j} + MFR + VAR + YEAR + DRAFT + \mu_{0j} + \varepsilon_{ij} \qquad (11)$$

The independent variables and fixed effects of level 3 are introduced on the basis of Eq (11), while other factors that have an impact on prices in problem 1 are introduced to improve the accuracy of the model.

## Multilevel modeling of regional effects

Referring to the study of Yale Camila Pareja et al. (2022) and other scholars, the analysis of price levels under different regions was achieved by introducing random intercepts and random slope coefficients [20,21]. We obtained the following results.

**Fixed effects results.** From the results in the Table 7, it can be obtained that the intercept term, *MFR*, *VAR*, *YEAR*, *DRAFT* all these independent variables show significance at 1% level in the model, which indicates that the overall effect of the model is good and there is a more obvious regional effect, i.e., there is a significant difference in the market price of used sailboats in different regions.

**Random effects results.** The analysis of random effects should be expressed using covariance, and the results are presented below.

According to the above Table 8, we find that the residual term is statistically significant, indicating that the variation in prices is clustered over different regions, i.e., there are individual differences. And it passes the significance test at the 1% level, indicating that the model has validity.

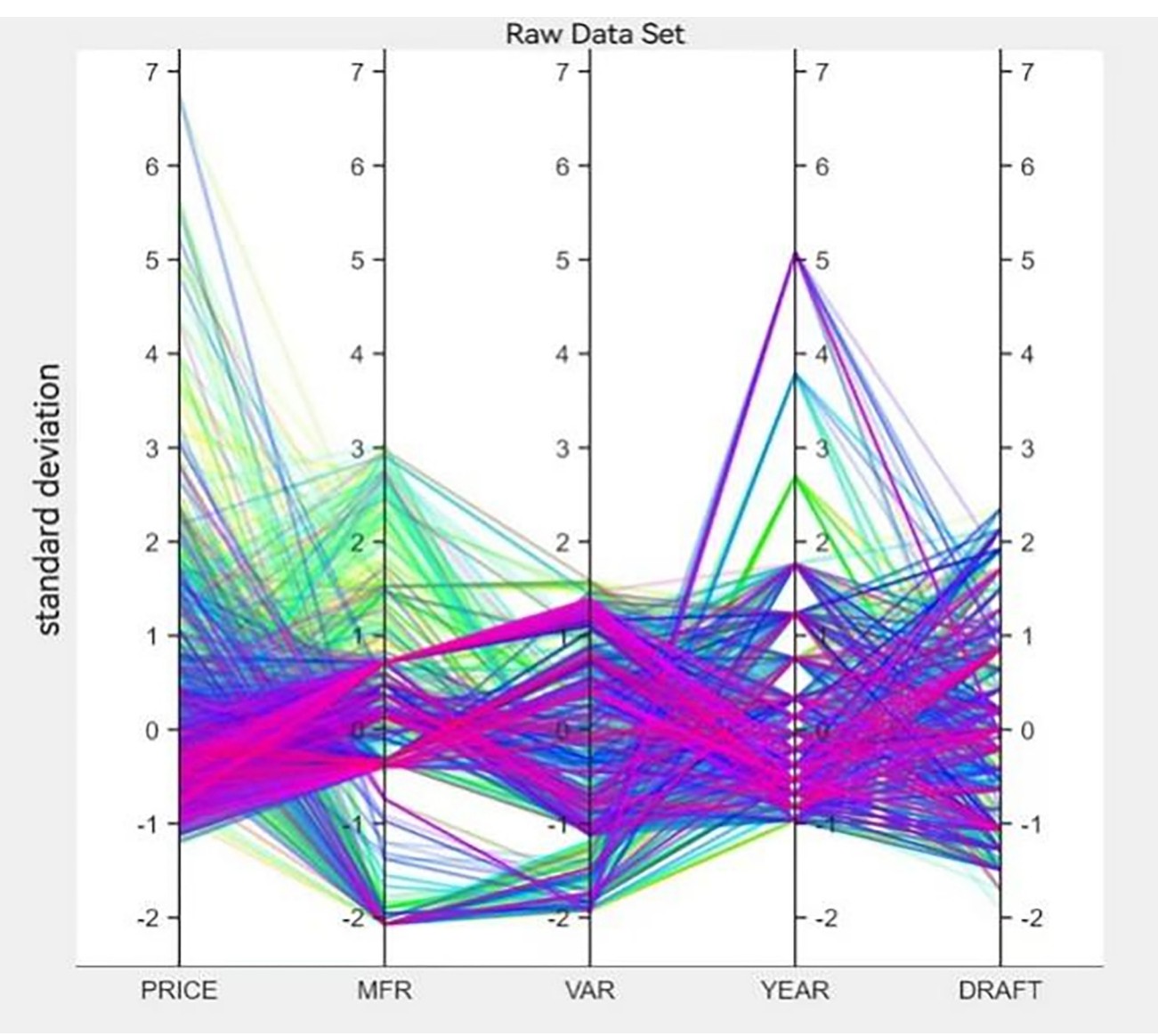

**Fig 14. Fine tree self-learning.**

## Consistency test based on fine tree model

To further assess the accuracy and consistency of the model, we used a modified refined tree model for prediction testing. We use 80% of the sample for machine self-learning and present

**Table 7. Type III test of fixed effects.**

| Source | Numerator df | Denominator df | F | P |
|---|---|---|---|---|
| Intercept term | 1 | 2811 | 675.932 | 0.000*** |
| MFR | 40 | 2811 | 12.376 | 0.000*** |
| VAR | 307 | 2811 | 4.624 | 0.000*** |
| YEAR | 14 | 2811 | 13.063 | 0.000*** |
| DRAFT | 25 | 2811 | 23.320 | 0.000*** |

**Table 8. Estimated values of covariance parameters.**

| Parameter | Estimated | Std. Error | Wald Z | P |
|---|---|---|---|---|
| Residual term | 1.04395E+10 | 278461457.5 | 37.490 | 0.000*** |

the following visualization of the fine tree classification results. The results are shown in Figs 14 and 15.

From Fig 14, we can see that the color of the data lines can be roughly divided into three categories, i.e., reddish, blueish, and greenish. Fig 15 shows the scatter plot of the relationship between *PRICE* and *MFR*. It can be seen that the conclusion of the fine tree machine self-learning is consistent with the conclusion obtained from the multilevel model of area effects.

To verify the accuracy of the model's machine self-learning, we perform a prediction test with the remaining 20% of the data sample. The results are shown in Figs 16 and 17.

According to Figs 16 and 17, the above prediction results show that the prediction results and the learning results generally remain consistent, with only certain differences in individual samples. The model has a more accurate effect and can be applied to the regional prediction and judgment of used sailboat prices, while the transferability of the multilevel model of regional effects makes it possible to extend it to the analysis of regional effects in other fields.

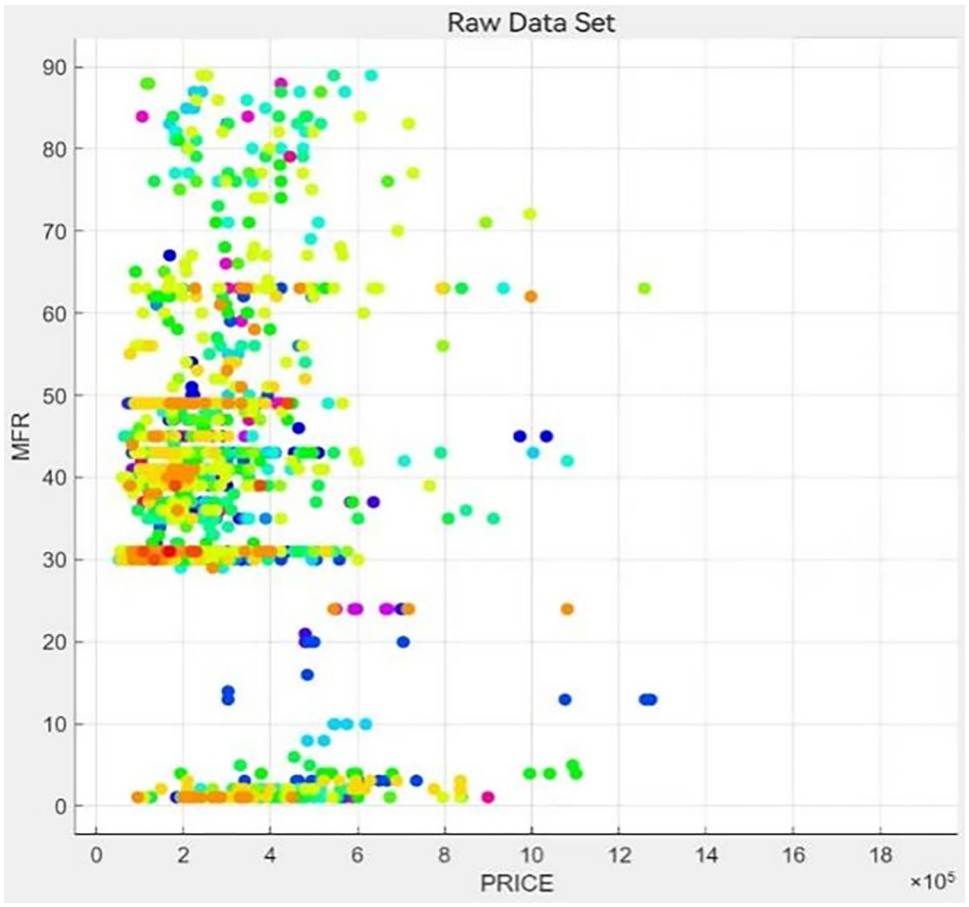

**Fig 15. Relationship scatter plot.**

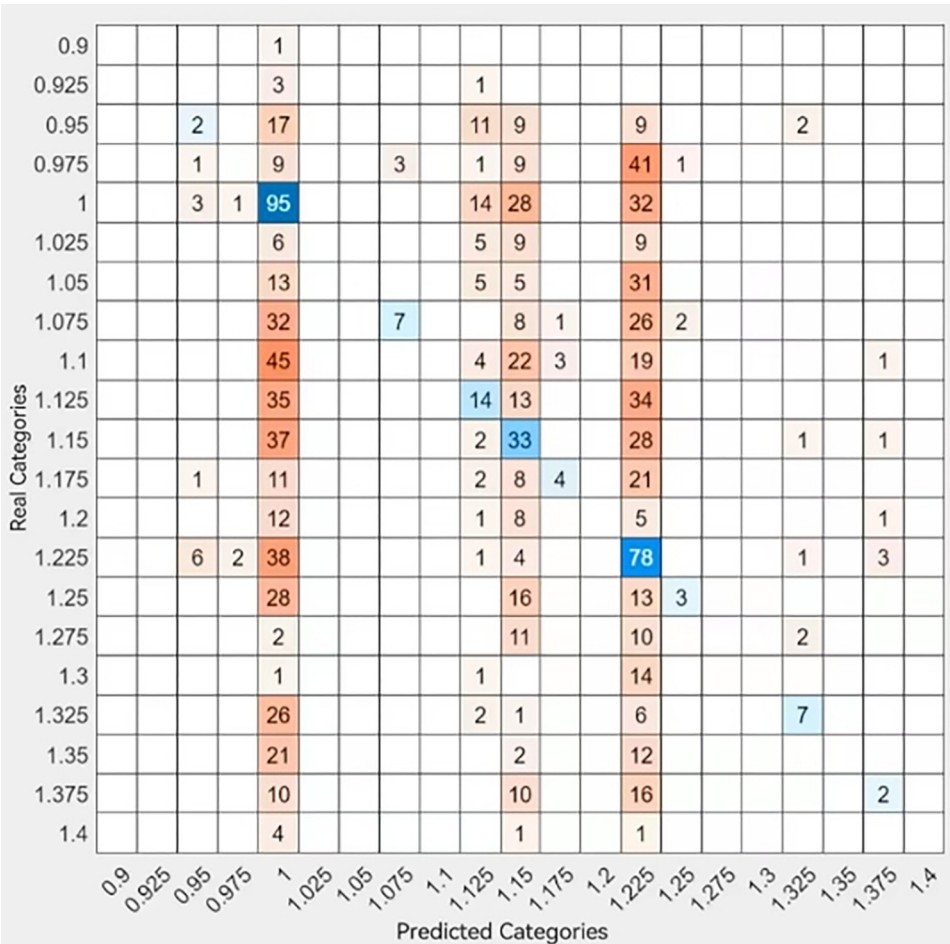

**Fig 16. Predicted effect.**

## Application examples on the Hong Kong market

According to the International Shipping Center Development Index (ISCD) 2018, Hong Kong is the second largest international shipping center in the world. Since used sailing ship prices have significant regional effects in Europe, USA, and Caribbean, then it is worthwhile to ponder whether the same phenomenon exists in the Hong Kong market. Therefore, we apply the above model to the Hong Kong market to test the generalizability of the model.

## NAR-MLM integrated model construction

**Comparable price measurement.** Due to the factors of price changes between years, current prices do not exactly reflect the dynamics of the physical economy [22]. Therefore, we use comparable prices to measure the valuation of second-hand sailboats in Hong Kong, and the more common accounting method is the price index deflator.

Price index deflator method

Use of price indices to eliminate the influence of price factors in dynamic indicators of product value volume. The formula is:

$$ComVA = \frac{CurVA}{BPPI} \tag{12}$$

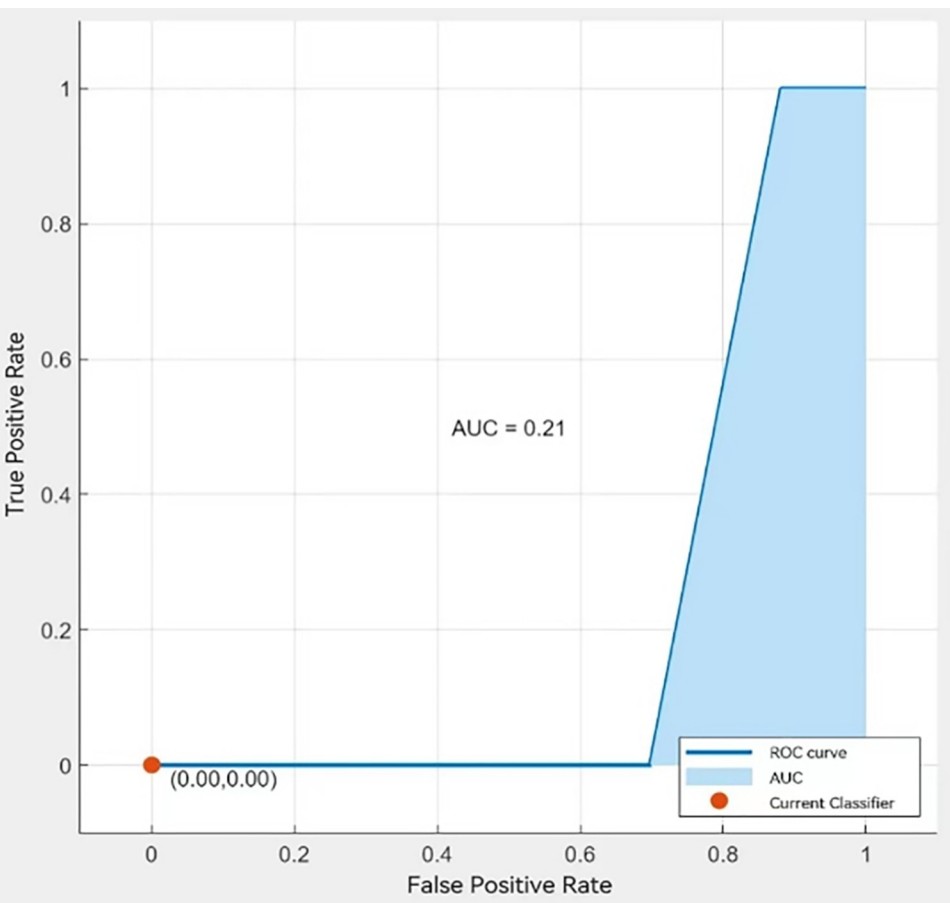

**Fig 17. AUC curve.**

Where, *ComVA* is the comparable price added value, *CurVA* is the current price added value, *BPPI* is the base period price index.

We have calculated comparable prices for the Hong Kong SAR market based on statistical data for the Hong Kong sailing industry, denoted as *ComPrice*. Data comes from *Census and Statistics Department* and *Hong Kong Monetary Authority*.

**Hong Kong market model.** Based on the two previous master models, we incorporate the Hong Kong comparable price *ComPrice* as a new indicator into the NAR dynamic neural network model, while additively introducing a stochastic intercept term on the regional effect to form the NAR-MLM dynamic model for the Hong Kong market. The model is constructed as follows.

$$\begin{cases} Y(t) = f(y(t-1), y(t-2), y(t-3), \cdots, y(t-d)) \\ y_{ij} = \alpha_0 + \alpha_{1j}x_{1ij} + \alpha_2 w_{2j} + \alpha_3 w_{3j} + MFR + VAR + YEAR + DRAFT + ComPrice + \mu_{0j} + \varepsilon_{ij} \end{cases} \quad (13)$$

## Comparative analysis of monohull and catamaran sailboats

We select the informative subset of sailboats from the dataset, divided into monohulls and catamarans, thus simulating the regional impact of Hong Kong (SAR) on the prices of monohull and catamaran sailboats in the subset.

**Comparison of evaluation results with different intercept terms.** In this paper, the evaluation results under different intercept items are compared, and the results are shown in Table 9.

**Table 9. Results of the assessment of the model intercept term.**

| Monohull Sailboats | | | | | Catamaran Sailboats | | | | |
|---|---|---|---|---|---|---|---|---|---|
| **Fixed effects** | **Coeff** | **S.E** | **t** | **P** | **Fixed effects** | **Coeff** | **S.E** | **t** | **P** |
| **Intercept** | 8.36 | 4.158 | 22.35 | <0.0000 | **Intercept** | 9.285 | 9.132 | 18.55 | <0.0000 |
| **Obs** | 2346 | 2346 | 2346 | 2346 | **Obs** | 1145 | 1145 | 1145 | 1145 |

**Table 10. Results of the impact of introducing area term.**

| Monohull Sailboats | | | | | Catamaran Sailboats | | | | |
|---|---|---|---|---|---|---|---|---|---|
| **Fixed effects** | **Coeff** | **S.E** | **t** | **P** | **Fixed effects** | **Coeff** | **S.E** | **t** | **P** |
| **Intercept** | -9.913 | 5.877 | -5.133 | <**0.0000** | **Intercept** | -5.845 | 4.523 | -0.293 | >0.1000 |
| **Area** | 2.254 | 4.356 | 5.313 | <**0.0000** | **Area** | 1.652 | 1.844 | 4.715 | <**0.0500** |
| **Obs** | 2346 | 2346 | 2346 | 2346 | **Obs** | 1145 | 1145 | 1145 | 1145 |

As can be seen from the data in the Table 9, both monohull and catamaran sailboats have good significance in the evaluation of the different intercept terms.

**Comparison of assessment results for different sail surface area items.** In this paper, the evaluation results of different sail surface area projects are compared, and the results are shown in Table 10.

As can be seen from the data in the Table 10, in the evaluation of different sail areas, monohull sailing boats all have a good significance, the effect of area of catamaran sailing boats is significant, while the effect of the intercept term is not significant.

## Other inferences and sensitivity analysis

### Spatial effect analysis

In addition, we analyze the relationship between price and regional differences among individuals by means of a spatial effects model.

As shown in Fig 18, there are some differences in the spatial effects between monohull and catamaran sailboats.

### Sensitivity analysis

In the sensitivity test, we perform an effect analysis of the NAR dynamic neural network prediction model based on the principle of gradient descent.

As seen in the above Fig 19, the NAR model can pass the gradient test, which indicates that the estimation of the model is good.

## Model evaluation

### Model advantages

In order to evaluate whether the model constructed in this study is applicable to the second-hand sailboat market and whether it has validity, accuracy and transferability, this section conducts a model evaluation to inform objective facts about the positive significance and limitations of the model.

**Good Performance:** this paper introduces dynamic factors based on the traditional neural network model and innovatively adopts a rolling NAR dynamic neural network model, which greatly improves the accuracy of the prediction of the secondary market pricing model.

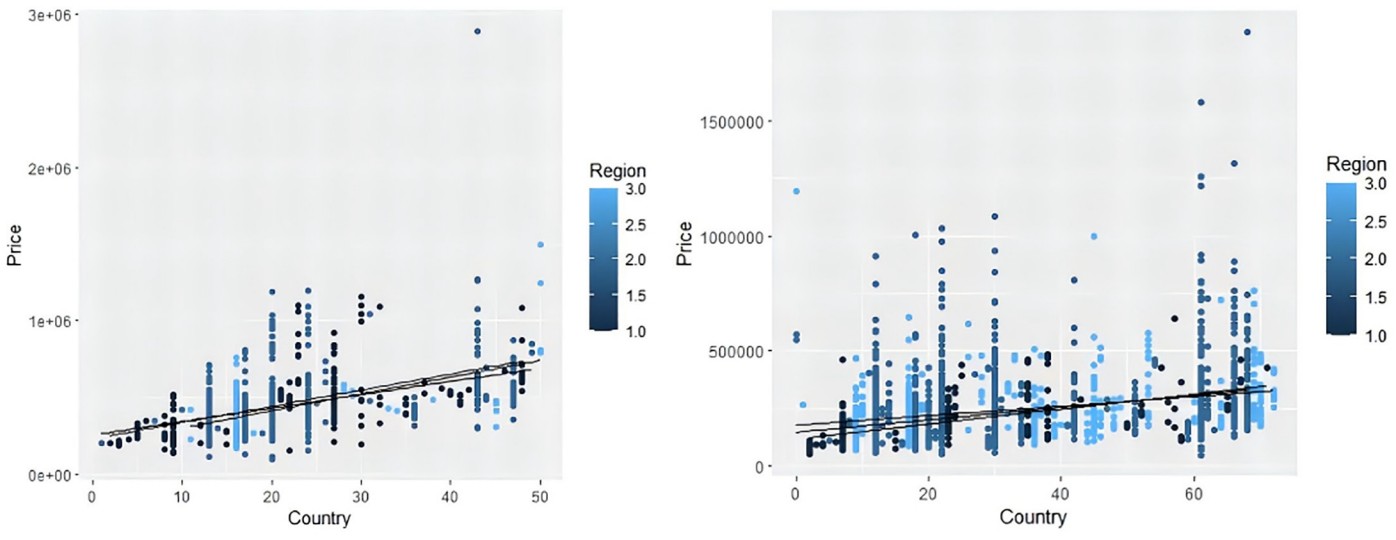

**Fig 18. Comparison of the spatial effects of monohull and catamaran sailboats.**

**Fig 19. NAR model effect map.**

Meanwhile, the novel multi-level model is used in the analysis of regional effects, which has good statistical effects.

**Effective Evaluation:** we construct the model based on the initial judgment of the data and conduct autocorrelation test, error analysis, consistency test and sensitivity analysis on the model effect to ensure the validity of the assessment.

**Transferability:** The main model in this paper has good consistency and generalizability for the NAR dynamic neural network pricing model and the regional effects multilevel model, which can be extended to price simulation and forecasting for various markets in different regions.

## Model weakness

**Simplified pricing mechanism:** Since the given data mainly relates to the macro environment and the indicators of the sailing category, micro-level influences are not considered, making the model relatively simplified.

## Conclusion

We developed a dynamic model of used sailboat pricing NAR to simulate pricing patterns and predict price trends. Subsequently, the impact of geographic regions was clarified by constructing a multi-level model of the regional effects of RE-MLM. In addition, based on market data from Hong Kong SAR, we accounted for comparable prices in Hong Kong and tested the feasibility and validity of the model using an integrated model analysis. Overall, this research contributes valuable insights to the secondhand economy, offering some practical references.

## Author Contributions

**Conceptualization:** Zhanni Huang, Di Wu.

**Data curation:** Zhanni Huang, Di Wu.

**Formal analysis:** Hansheng Hu.

**Methodology:** Zhanni Huang.

**Resources:** Hansheng Hu, Di Wu.

**Supervision:** Di Wu.

**Writing – original draft:** Zhanni Huang, Hansheng Hu, Di Wu.

**Writing – review & editing:** Zhanni Huang.

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
