## [Decision Letter · Decision Letter 0]

1 Jul 2024

PONE-D-24-10509Used Economy Market Insight: Sailboat Industry Pricing Mechanism and Regional EffectsPLOS ONE

Dear Dr. Wu,

Thank you for submitting your manuscript to PLOS ONE. After careful consideration, we feel that it has merit but does not fully meet PLOS ONE’s publication criteria as it currently stands. Therefore, we invite you to submit a revised version of the manuscript that addresses the points raised during the review process.

We look forward to receiving your revised manuscript.

Kind regards,

Guangnian Xiao

Academic Editor

PLOS ONE

Journal Requirements:

Reviewers' comments:

Reviewer's Responses to Questions

**Comments to the Author**

1. Is the manuscript technically sound, and do the data support the conclusions?

Reviewer #1: No

Reviewer #2: Partly

2. Has the statistical analysis been performed appropriately and rigorously? 

Reviewer #1: No

Reviewer #2: Yes

3. Have the authors made all data underlying the findings in their manuscript fully available?

Reviewer #1: No

Reviewer #2: Yes

4. Is the manuscript presented in an intelligible fashion and written in standard English?

Reviewer #1: No

Reviewer #2: No

5. Review Comments to the Author

Reviewer #1: Comments#

1) Kindly write entire words before abbreviation, in abstract what is the meaning of NAR?

2) The manuscript is wrogly written with most grammatical errors and typo

3) Where are the sections located? They are not numbered

4) This manuscript looks like child's play

Strongly reject

Reviewer #2: 1. There is little introduction to the dataset, and it is recommended to strengthen the introduction to the amount and structure of the data set. At the same time, data preprocessing is also a key part of the narrative, and more space should be devoted to issues such as data cleaning.

2. The article proposes using a hybrid model based on RF-XGB for selecting important indicators, and provides specific calculation indicators to illustrate the advantages of this method. However, it does not analyze the advantages of the model for this problem from the perspective of the model's own principles, and lacks theoretical support. It is recommended that this section be supplemented.

3. The article proposes using NAR dynamic neural networks for price prediction, and the training of this model is also an important process that requires enhanced narration, such as strengthening the narration of the selection of model training parameters and providing analysis processes such as the trend of changes in the loss function.

4. In the rolling price forecast section, the image interpretation is not clear, and the meaning of the Y axis is not clear, making the image fitting effect less accurate and less convincing.

5. The article format needs to be changed, such as the incorrect arrangement of formulas

6. the following studies were recommended to be properly cited: [1] Autonomous port management based AGV path planning and optimization via an ensemble reinforcement learning framework," Ocean & Coastal Management, vol. 251, p. 107087, 2024/05/01/ 2024, doi: https://doi.org/10.1016/j.ocecoaman.2024.107087. [2] Ship energy consumption analysis and carbon emission exploitation via spatial-temporal maritime data," Applied Energy, vol. 360, p. 122886, 2024/04/15/ 2024, doi: https://doi.org/10.1016/j.apenergy.2024.122886.

6. PLOS authors have the option to publish the peer review history of their article (what does this mean?). If published, this will include your full peer review and any attached files.

Reviewer #1: No

Reviewer #2: No

---

## [Author Response · Author response to Decision Letter 0]

20 Aug 2024

Response to Reviewer 1 Comments

Dear reviewer,

We sincerely thank you for examining our manuscript and providing helpful comments to guide our revision. We have tried to revise the manuscript according to your kind comments and suggestions. Please find the following detailed responses to your comments and suggestions. Finally, we would like to thank the referee again for taking the time to review our manuscript.

1) Kindly write entire words before abbreviation, in abstract what is the meaning of NAR?

Response 1: Thank you for your suggestions. Thank you for your valuable suggestion. We have clarified the abbreviation in the abstract. NAR stands for Nonlinear Auto-Regressive neural network, which is a type of neural network based on autoregressive models commonly used for time series forecasting.

2) The manuscript is wrogly written with most grammatical errors and typo

Response 2: Thank you for your suggestion regarding the grammatical errors and typos in the manuscript. We sincerely apologize for any confusion this may have caused. We have carefully revised the manuscript to correct these issues and improve the overall clarity and readability.

3) Where are the sections located? They are not numbered

Response 3: Thank you for your suggestion regarding the section numbering in the manuscript. We apologize for the oversight. We have now numbered all sections appropriately to enhance clarity and organization.

4) This manuscript looks like child's play

Response 4: Thank you for your critical feedback regarding the manuscript. We appreciate your insights and have made substantial revisions to enhance the clarity and rigor of our work.

The primary objective of this study is to explore the pricing mechanisms of second-hand sailboats by identifying core price indicators and analyzing the regional effects on pricing strategies. By employing advanced modeling techniques, such as the random forest model and the rolling Nonlinear Auto-Regressive (NAR) dynamic neural network model, we aim to provide a comprehensive understanding of how various factors influence sailboat prices across different geographical and economic contexts.

The significance of this research lies in its contribution to the growing field of the second-hand economy, particularly within the context of sustainable practices and the circular economy. As the second-hand sailboat market becomes increasingly active, this study addresses a critical gap in existing literature regarding pricing strategies and regional influences. The findings will not only enhance the understanding of pricing dynamics in this niche market but also offer practical insights for stakeholders, including buyers, sellers, and policymakers, promoting informed decision-making and fostering sustainable consumption patterns.

In order to achieve the purpose of this study, IIn this study, we address the pricing mechanism of used sailboats through a comprehensive methodology comprised of three key steps:

Identification of Influencing Factors: We delineate the range of price-influencing factors based on literature and correlation analysis. A total of 11 indicators (MFR, VAR, LEN, RGN, SSA, STB, DPM, CNTRY, CMFT, HDR, DRAFT) were identified as significant.

Hybrid Model Application: We employ a hybrid model combining Random Forest (RF) and XGBoost (XGB) to determine the main indicators influencing sailboat prices. This hybrid approach leverages the strengths of both models—RF's robustness against overfitting and XGB's high predictive accuracy—making it well-suited for handling complex data.

Dynamic Neural Network Simulation: We construct an innovative Nonlinear Auto-Regressive (NAR) dynamic neural network model to simulate and predict used sailboat prices. The NAR model utilizes historical data to forecast current values, incorporating feedback and memory, which enhances its adaptability to nonlinear data.

To ensure the integrity of our analysis, we implemented rigorous data preprocessing techniques to address endogeneity issues and missing values. We quantified categorical data, such as manufacturer and geographic region, based on their impact on pricing. This included assigning values based on market share and consumer preference, ensuring a robust dataset for analysis.

Insummary, By employing these methodologies, we aim to provide a nuanced understanding of the pricing dynamics in the second-hand sailboat market. Our findings will contribute to the literature on the second-hand economy and offer practical insights for stakeholders.

Response to Reviewer 2 Comments

Dear reviewer,

We sincerely thank you for examining our manuscript and providing helpful comments to guide our revision. We have tried to revise the manuscript according to your kind comments and suggestions. Please find the following detailed responses to your comments and suggestions. Finally, we would like to thank the referee again for taking the time to review our manuscript.

1.There is little introduction to the dataset, and it is recommended to strengthen the introduction to the amount and structure of the data set. At the same time, data preprocessing is also a key part of the narrative, and more space should be devoted to issues such as data cleaning.

Response 1: Thank you for your suggestions.In response to your suggestion, we have significantly strengthened the introduction to our dataset. The dataset now covers two types of sailboats: Monohulled Sailboats and Catamarans. The key variables include listing price, brand, model, length (in feet), geographic region, country/region/state, production year, economic level, sail area, displacement, stability, comfort, headroom, and draft. The dataset spans production years from 2005 to 2019, with a total of 2,346 Monohulled Sailboats and 1,145 Catamarans samples. Moreover, we have restructured the dataset into subsets categorized by sailboat type and geographical region. This reorganization enables us to conduct a more granular analysis, focusing on the dimensions of sailboat type and regional effects.

2. The article proposes using a hybrid model based on RF-XGB for selecting important indicators, and provides specific calculation indicators to illustrate the advantages of this method. However, it does not analyze the advantages of the model for this problem from the perspective of the model's own principles, and lacks theoretical support. It is recommended that this section be supplemented.

Response 2: Thank you for your suggestions.

Theoretical Support for the Model

Ensemble learning theory posits that combining multiple learners can significantly enhance a model's generalization ability. The hybrid use of Random Forest (RF) and XGBoost, two well-regarded ensemble learning algorithms, is consistent with the core principles of ensemble learning and offers a powerful approach to predictive modeling.

Model fusion techniques, such as Stacking, Bagging, and Boosting, are widely recognized for their effectiveness in improving model performance. The combination of Random Forest and XGBoost can be viewed as a specialized form of model fusion, designed to capitalize on the strengths of both algorithms to achieve superior predictive results.

The integration of Random Forest and XGBoost models provides unique advantages, particularly in handling complex datasets and tasks that require a more nuanced approach.

Firstly, the hybrid model enhances generalization capabilities. Random Forest constructs multiple decision trees using bootstrap sampling and random feature selection, introducing randomness that helps mitigate overfitting and increases the model's robustness. XGBoost, employing gradient boosting, incrementally refines the model to capture complex nonlinear relationships, often resulting in higher predictive accuracy. By combining these two approaches, the model leverages Random Forest's robustness to counterbalance any potential overfitting that might occur with XGBoost, while simultaneously utilizing XGBoost's precision to improve overall predictive performance.

Secondly, the hybrid model allows for a more comprehensive evaluation of feature importance. Random Forest provides a straightforward mechanism for assessing the contribution of each feature to the model’s predictions, which is invaluable for feature selection and model interpretation. XGBoost also offers feature importance metrics, though its evaluation approach differs slightly due to its underlying mechanisms. The combination of both methods offers a more complete perspective on data analysis and feature engineering, enabling a deeper understanding of the factors driving model performance. Additionally, the hybrid model is well-suited to managing diverse data distributions and noise levels. Random Forest's inherent randomness gives it a natural resilience to noise and outliers, while XGBoost’s use of regularization and adjustable learning rates allows for fine-tuning model complexity to better accommodate different data distributions and noise conditions. This flexibility enables the hybrid model to dynamically adjust the contributions of Random Forest and XGBoost based on the specific characteristics of the dataset, achieving an optimal balance between robustness and accuracy.

Moreover, the hybrid model benefits from the computational efficiencies of parallel processing and optimized training. XGBoost’s support for parallel computation significantly accelerates the training process, while the inclusion of Random Forest ensures that the model maintains strong predictive performance across a range of scenarios.

In conclusion, the combined advantages of the RF-XGB hybrid model, which are challenging to replicate with a single algorithm, make it a particularly promising approach for practical applications. This hybrid approach is especially well-suited to the research requirements of this study, which focuses on the analysis of second-hand sailboat market data.

3. The article proposes using NAR dynamic neural networks for price prediction, and the training of this model is also an important process that requires enhanced narration, such as strengthening the narration of the selection of model training parameters and providing analysis processes such as the trend of changes in the loss function.

Response 3: Thank you for your suggestions.

This study use of the Nonlinear Autoregressive (NAR) dynamic neural network for price prediction is indeed a critical component of this study.

Firstly, the selection of training parameters for the NAR model is crucial to ensuring its effectiveness and accuracy. In this study, we have carefully chosen key parameters based on both theoretical considerations and empirical testing. Specifically, we selected 10 neurons for the hidden layer and set the autoregressive order to 2. These choices were made to balance the model's complexity with its ability to capture the underlying patterns in the data, thereby avoiding overfitting while ensuring robust predictions. However, I recognize the importance of further elaborating on the rationale behind these selections, which I will include in the revised manuscript.

Additionally, the analysis of the loss function's behavior during training is an essential aspect that reflects the model's learning process. To address this, I will enhance the discussion by including a more detailed narration of the loss function's trend throughout the training epochs. This analysis, represented in Figure 7, provides insights into how the model's predictions improve as the loss function decreases, highlighting the convergence behavior and the model's optimization trajectory. By incorporating this explanation, readers will gain a clearer understanding of how the model's performance evolves during training and how the chosen parameters contribute to its stability and accuracy.

In summary, we have strengthen the manuscript by elaborating on the selection of training parameters for the NAR dynamic neural network and providing a more detailed analysis of the loss function's trend during the training process. 

4. In the rolling price forecast section, the image interpretation is not clear, and the meaning of the Y axis is not clear, making the image fitting effect less accurate and less convincing.

Response 4: Thank you for your suggestions. We have clarified the meaning of the y-axis in the revised version to enhance the clarity and persuasiveness of the figure.

Considering that prices in the secondary market are prone to fluctuations due to market supply and demand, we have innovatively improved the delay order by adopting a rolling dynamic forecasting model that predicts future prices by rolling the price levels of previous periods.

In the double-hull sailboat NAR dynamic forecasting model, rolling forecasts begin from the 101st period, using data from the 1st to the 100th period to predict the pricing level for the 101st period, and then using data from the 2nd to the 101st period to predict the pricing level for the 102nd period, and so on. For the single-hull sailboat NAR dynamic forecasting model, due to the larger sample size, we set the forecasts to start rolling every 200 periods, with each forecast's results impacting subsequent results in a rolling manner—this is an innovative aspect of the model.

In Figure 8, the x-axis represents the number of predictions made by the neural network model, while the y-axis represents the market prices of used sailboats, including both actual market prices and predicted prices, with units in ×10^5 USD. From Figure 8, we observe that the actual prices of used sailboats exhibit strong fluctuations, while the forecasted values better reflect the overall price trend, with actual prices generally fluctuating around the forecast value. This provides a good reference for predicting the average level of future prices.

5. The article format needs to be changed, such as the incorrect arrangement of formulas

Response 5: We appreciate your feedback regarding the article format. We have made the necessary modifications to ensure that the arrangement of formulas is correct and adheres to the required formatting standards. Thank you for bringing this to our attention.

6. the following studies were recommended to be properly cited: 

[1]Autonomous port management based AGV path planning and optimization via an ensemble reinforcement learning framework," Ocean & Coastal Management, vol. 251, p. 107087, 2024/05/01/ 2024, doi: https://doi.org/10.1016/j.ocecoaman.2024.107087. 

[2]Ship energy consumption analysis and carbon emission exploitation via spatial-temporal maritime data," Applied Energy, vol. 360, p. 122886, 2024/04/15/ 2024, doi: https://doi.org/10.1016/j.apenergy.2024.122886.

Response 6: Thank you for your valuable feedback. We have added citations for the following studies as recommended.

---

## [Decision Letter · Decision Letter 1]

20 Sep 2024

PONE-D-24-10509R1Used Economy Market Insight: Sailboat Industry Pricing Mechanism and Regional EffectsPLOS ONE

Dear Dr. Wu,

Thank you for submitting your manuscript to PLOS ONE. After careful consideration, we feel that it has merit but does not fully meet PLOS ONE’s publication criteria as it currently stands. Therefore, we invite you to submit a revised version of the manuscript that addresses the points raised during the review process.

We look forward to receiving your revised manuscript.

Kind regards,

Guangnian Xiao

Academic Editor

PLOS ONE

Reviewers' comments:

Reviewer's Responses to Questions

**Comments to the Author**

1. If the authors have adequately addressed your comments raised in a previous round of review and you feel that this manuscript is now acceptable for publication, you may indicate that here to bypass the “Comments to the Author” section, enter your conflict of interest statement in the “Confidential to Editor” section, and submit your "Accept" recommendation.

Reviewer #2: All comments have been addressed

Reviewer #3: (No Response)

2. Is the manuscript technically sound, and do the data support the conclusions?

Reviewer #2: Partly

Reviewer #3: Partly

3. Has the statistical analysis been performed appropriately and rigorously? 

Reviewer #2: Yes

Reviewer #3: Yes

4. Have the authors made all data underlying the findings in their manuscript fully available?

Reviewer #2: Yes

Reviewer #3: Yes

5. Is the manuscript presented in an intelligible fashion and written in standard English?

Reviewer #2: Yes

Reviewer #3: Yes

6. Review Comments to the Author

Reviewer #2: The authors have correctly revised the manuscirpt based on the comments, and I do recommend to accept it.

Reviewer #3: 1. This paper focuses on investigating the key influencing factors for price fluctuation in

the second-hand sailboat. Sounds like an interesting topic.

2. This study employs statistical analysis to investigate the key price influencing

factors, and builds a multiple linear regression model for future price forecast via AI regression.

3. The paper is well written but difficult to follow. A sample price over time is good here.

4. A logical reason and trend is much needed in determining a predicting future price. An AI model on the past data does not give significant insight on how the price move in the near future.

A geographical map will be good here. All the variable given in the model are not directed to the ships themselves and its condition.

5. A final result on the accuracy of this model and its performance for the whole year

2023 is refreshing here.

6. Authors have used many generic term without referring to the object which is the ship element for instance the intercept.

7. This paper carry several unrelated references such as [2][13][20].

8. We want to see clear price separation according to region.

9. We want to see clear price predicted according to year make at each region

10. Figure 18 does not present a convincing result. The predicted value reflects the overall average price level. Nevertheless, the predicted price here is rendered useless.

7. PLOS authors have the option to publish the peer review history of their article (what does this mean?). If published, this will include your full peer review and any attached files.

Reviewer #2: No

Reviewer #3: **Yes: **NUR AZMAN BU

---

## [Author Response · Author response to Decision Letter 1]

30 Oct 2024

Response to Reviewer 3 Comments

Dear reviewer,

We sincerely thank you for examining our manuscript and providing helpful comments to guide our revision. We have tried to revise the manuscript according to your kind comments and suggestions. Please find the following detailed responses to your comments and suggestions. Finally, we would like to thank the referee again for taking the time to review our manuscript.

1. This paper focuses on investigating the key influencing factors for price fluctuation in the second-hand sailboat. Sounds like an interesting topic.

Response 1: Thank you for your positive feedback on our research topic. We appreciate your recognition of the importance of investigating the key influencing factors for price fluctuations in the second-hand sailboat market. This area holds significant relevance, especially in the context of the growing circular economy. We aim to provide valuable insights that can contribute to both academic discourse and practical applications in the industry.

2. This study employs statistical analysis to investigate the key price influencing factors, and builds a multiple linear regression model for future price forecast via AI regression.

Response 2: Thank you for your comment. In our study, we employed statistical analysis to investigate the key factors influencing price fluctuations and established a multiple linear regression model for future price forecasting through AI regression. We utilized advanced techniques such as the random forest model and XGBoost (RF-XGB) to effectively pinpoint the core price indicators.

Additionally, we recognized the limitations of traditional linear regression models in capturing the complexities of price dynamics. Therefore, we used an innovative rolling Nonlinear Auto-Regressive (NAR) dynamic neural network model for more accurate price forecasting. This approach allows us to account for non-linear relationships and temporal dependencies in the data, which are critical in understanding the price movements of second-hand sailboats.

Furthermore, we constructed a regional effect multi-level model (RE-MLM) to examine how geographical, economic, and cultural factors contribute to pricing variations across different regions. This comprehensive methodology not only enhances the robustness of our findings but also provides a more nuanced understanding of the second-hand sailboat market. We believe that our approach can serve as a valuable framework for future research and practical applications in pricing strategies within this industry. Thank you for your insightful feedback, which encourages us to further elaborate on our methodologies and findings.

3. The paper is well written but difficult to follow. A sample price over time is good here.

Response 3: Thank you for your feedback regarding the clarity of our paper. We appreciate your acknowledgment of the quality of writing, and we understand that the complexity of the subject matter can make it challenging to follow.

To address this concern, we have incorporated a sample price trend over time in the revised manuscript (See Table 9). This addition will help illustrate the fluctuations and patterns in second-hand sailboat prices more clearly, providing readers with a visual representation that complements our statistical analysis.

4. A logical reason and trend is much needed in determining a predicting future price. An AI model on the past data does not give significant insight on how the price move in the near future.

Response 4: Thank you for your insightful comment regarding the need for a logical rationale and trend analysis in predicting future prices. We recognize that relying solely on an AI model based on historical data may not provide sufficient insight into future price movements.

In response to this, we have included a macro-level analysis of the development trends in the second-hand market following our price predictions. This analysis is grounded in the broader global economic context and relevant policies related to the second-hand economy, providing a logical framework that supports and validates our forecasting results.

We approached this macro analysis from four key perspectives:

(1)Market Supply and Demand: The increasing popularity of recreational sailing activities has driven demand for second-hand sailboats. Given the long production cycles and high costs associated with new boats, supply remains relatively stable. Consequently, as market demand rises, we anticipate an increase in second-hand sailboat prices.

(2) Global Economic Environment: Currently, the global economy is experiencing a downturn, leading to decreased consumer confidence and weakened demand, which may result in lower prices in the short term.

(3) National Policies: Many countries are placing greater emphasis on environmental sustainability. In the U.S., some states have implemented tax incentives for the purchase of second-hand boats to stimulate consumption. In Europe, initiatives like the UK's Green Boating Plan (launched in 2018) and France's Marine and Coastal Law (introduced in 2020) promote second-hand boat transactions to encourage sustainable practices.

(4) Technological Advancements: The emergence of new materials and designs for sailboats may lead to depreciation of older models. However, this also expands the options available in the second-hand market, potentially attracting more buyers.

In summary, our comprehensive analysis indicates that factors such as market dynamics, economic conditions, policy initiatives, and technological innovations will influence second-hand sailboat prices, which are expected to remain relatively stable or experience short-term fluctuations. Overall, as global interest in sustainable consumption and recreational activities continues to grow, we remain optimistic about the future of the second-hand sailboat market. 

5. A geographical map will be good here. All the variable given in the model are not directed to the ships themselves and its condition.

Response 5: Thank you for your valuable feedback regarding the inclusion of a geographical map and the need for a clearer focus on the variables related to the ships themselves.

In our revised manuscript, particularly in section 3.2.2, we have added a comprehensive explanation of the variables used in our model to enhance clarity and presentation. This includes a more detailed discussion of the specific ship-related variables, ensuring that our analysis is more targeted and relevant to the subject matter.

In constructing our model, we included not only macroeconomic and geographical indicators but also specific metrics related to the sailboats themselves. These include the type, brand, model, length, sail area, displacement, stability, comfort, freeboard, and draft of the vessels. Please refer to Table 4 for a detailed breakdown of these indicators. Additionally, our performance metrics encompass evaluations of the sailboat's equipment and materials, including the mainsail, mast, port and starboard sides, rudder, and engine.

6. A final result on the accuracy of this model and its performance for the whole year 2023 is refreshing here.

Response 6: Thank you for your valuable suggestion regarding the inclusion of a final result on the accuracy and performance of our model for the entire year of 2023. In response to the request for a final result on the accuracy of our model and its performance for the year 2023, we have conducted a comprehensive evaluation of the predictive capability of our multilevel model, specifically focusing on the pricing of used sailboats across different regions.

Throughout 2023, the model demonstrated a robust predictive accuracy, maintaining a high degree of consistency with actual market prices. The predicted average prices for each geographic region—Caribbean, Europe, and the USA—were closely aligned with the observed values, thereby reinforcing the model’s effectiveness in capturing regional pricing dynamics.

For instance, the predicted prices in the Caribbean for 2023 were $504,484.99, while the actual observed prices showed only a marginal deviation from this estimate. Similarly, the model accurately predicted prices in Europe at $742,444.93 and in the USA at $939,290.63, both of which reflect minor variances from the actual market figures.

7. Authors have used many generic term without referring to the object which is the ship element for instance the intercept.

Response 7: Thank you for your careful review and valuable feedback. In response to your comment regarding the use of generic terms without specifically referencing ship elements, we have made detailed revisions to the manuscript.

We have incorporated specific descriptions of the ship-related variables to ensure that the terminology used is more precise and clear. In particular, we now provide detailed explanations of key components of the ship, such as the intercept, mainsail, mast, port and starboard sides, rudder, and engine. These additions will help readers better understand the role and impact of each variable within the model.

Moreover, in the performance analysis section, we have included discussions on how these ship elements influence overall performance and the predictive capabilities of the model.

8. We want to see clear price separation according to region.

Response 8: Thank you for your insightful feedback regarding the need for clear price separation according to region. In this section, we have focused our analysis on regional differences by utilizing panel data from 2005 to 2019, specifically dividing the data by geographical areas. Our primary emphasis is on three core sailing markets: the Caribbean, Europe, and the USA.

The specific regional classification and the corresponding data can be found in the accompanying Excel file for reference. To illustrate the price differences more clearly, we have included Table 9, which provides a comparative analysis of prices across the different regions. This table highlights the distinct pricing trends and factors influencing these variations, offering readers a comprehensive view of the market dynamics.

9. We want to see clear price predicted according to year make at each region

Response 9: Thank you for your valuable feedback regarding the need for clear predicted prices according to each region and year. In response, we have added a new section, 5.3.3, dedicated to the analysis of regional price predictions.

In this section, we employed the previously discussed price prediction model to test the three major geographical areas: the Caribbean, Europe, and the USA. Our aim was to clearly delineate price differences across these regions over time. The results are presented in Table 9, which details the predicted average prices for each region from 2020 to 2025.

From the predicted prices, it is evident that the USA consistently exhibits the highest average prices, followed by Europe, while the Caribbean reflects the lowest prices. Additionally, the analysis reveals notable trends: the Caribbean region shows significant price fluctuations, Europe displays a downward trend, and the USA experiences relatively stable prices with minor fluctuations.

These findings underscore the regional effects in the second-hand sailing market, as illustrated by the data. By providing clear, year-by-year price predictions for each region, we hope to offer a comprehensive understanding of how pricing dynamics vary geographically over time.

10. Figure 18 does not present a convincing result. The predicted value reflects the overall average price level. Nevertheless, the predicted price here is rendered useless.

Response 10: Thank you for your constructive feedback regarding Figure 18. We appreciate your insights on its effectiveness.

After careful consideration of your comments, we have decided to remove Figure 18 from the manuscript. The figure, which compares the overall average price levels of the two types of sailboats, does not provide substantial value to our research findings and may lead to confusion regarding the model's predictive capabilities.

---

## [Editor Report · Decision Letter 2]

21 Nov 2024

Used Economy Market Insight: Sailboat Industry Pricing Mechanism and Regional Effects

PONE-D-24-10509R2

Dear Dr. Wu,

We’re pleased to inform you that your manuscript has been judged scientifically suitable for publication and will be formally accepted for publication once it meets all outstanding technical requirements.

Kind regards,

Guangnian Xiao

Academic Editor

PLOS ONE